# FRAME—Monte Carlo model for evaluation of the stable isotope mixing and fractionation

**Maciej P. Lewicki** [1] *, **Dominika Lewicka-Szczebak**[2], **Grzegorz Skrzypek**[3]

**1** Institute of Nuclear Physics, Polish Academy of Sciences, Krakow, Poland, **2** Institute of Geological Sciences, University of Wrocław, Wrocław, Poland, **3** West Australian Biogeochemistry Centre, School of Biological Sciences, The University of Western Australia, Perth, Australia

* maciej.lewicki@ifj.edu.pl

## Abstract

Bayesian stable isotope mixing models are widely used in geochemical and ecological studies for partitioning sources that contribute to various mixtures. However, none of the existing tools allows accounting for the influence of processes other than mixing, especially stable isotope fractionation. Bridging this gap, new software for the stable isotope Fractionation And Mixing Evaluation (FRAME) has been developed with a user-friendly graphical interface (malewick.github.io/frame). This calculation tool allows simultaneous sources partitioning and fractionation progress determination based on the stable isotope composition of sources/substrates and mixture/products. The mathematical algorithm applies the Markov-Chain Monte Carlo model to estimate the contribution of individual sources and processes, as well as the probability distributions of the calculated results. The performance of FRAME was comprehensively tested and practical applications of this modelling tool are presented with simple theoretical examples and stable isotope case studies for nitrates, nitrites, water and nitrous oxide. The open mathematical design, featuring custom distributions of source isotope signatures, allows for the implementation of additional processes that alternate the characteristics of the final mixture and its application for various range of studies.

## 1 Introduction

Partitioning of sources contributing to the stable isotope composition of a mixture, the *isotope mixing*, also known as a *stable isotope mass balance*, enable determination of $n + 1$ source contributions for $n$ isotopes analysed in a sample. This purely algebraic calculation technique has been employed for a long time across various disciplines. However, we often deal with more than $n + 1$ sources, that require application of iterative calculations, such as Bayesian statistics, to obtain the probability distributions of possible outcomes [1–3]. Such methods will provide solutions for complex cases that cannot be solved using algebraic calculations; however, they will not provide an exact result but a range of statistically probable solutions.

Stable isotope mixing models were initially developed for food-web studies to identify dietary proportions [2, 4–7]. These models usually allow distinguishing more than $n + 1$ sources for $n$ analysed isotopes. The model output provides the probability distributions of source contributions, including uncertainty, respectively to uncertainties arising from measurements and

**Data Availability Statement:** All the data can be found in the manuscript. All the code is accessible at web repository: github.com/malewick/frame.

**Funding:** GS: Coordinated Research Project of IAEA D15018, International Atomic Energy Agency,

url:https://www.iaea.org/projects/crp/d15018, NO DSL(1): internal grant of the University of Wrocław (10120, https://uni.wroc.pl/) DSL(2): 3205/2003/20, Polish National Agency for Academic Exchange (by the Program 'Polish Returns'), https://nawa.gov.pl/en/ ML: author has received no specific funding for this work The funders had no role in study design, data collection and analysis, decision to publish, or preparation of the manuscript.

**Competing interests:** The authors have declared that no competing interests exist.

determination of source isotope signatures. This uncertainty is usually reported as a confidence interval of the contributing fractions. These models have already been adopted for defining water, dust, organic matter, or aerosol origins [8–10] and are also commonly used for the partitioning of nitrate sources in groundwaters [11–13] or surface waters [14–16].

However, the existing tools for applying isotope mixing models do not include the possible further isotopic fractionation of the product after mixing. This is usually not critical for food-web studies, where fixed values for fractionation factors are used, but is often important for other stable isotope analyses. For example, nitrates may undergo intensive denitrification processes, which can significantly alter their final isotopic signatures [17–19]. Therefore, the application of a simple mixing model neglecting possible further isotopic fractionation of the final product may result in a bias in the model outputs [11, 13–16, 20]. Some studies applying the nitrate mixing model manage to partially include denitrification fractionation in the model as a parameter added to the source isotopic signatures but without taking into account the fractionation progress [12]. This is a simplification assuming that the amount of reduced nitrate is constant across measured samples and this assumption results in a simple linear offset in the residual nitrate isotopic signature. However, this approach is not sufficient for a precise description of mixture origins, if significant fractionation occurs and determination of the fractionation progress is required.

Precise accounting for the fractionation progress is vital for the determination of $N_2O$ origins, that requires simultaneous determination of the fractions of the $N_2O$ originating from different production pathways and the fraction of reduced $N_2O$ [21, 22]. Hence, the mixing model must be combined with the potential isotopic fractionation of the produced $N_2O$ during its reduction to $N_2$ [22]. $N_2O$ is a very unique compound that can be analysed to obtain three different stable isotope signatures: $\delta^{15}N$, $\delta^{18}O$ and SP (site preference). SP provides additional information reflecting the $^{15}N$ enrichment in the central N position of the linear $N_2O$ molecule in relation to the external N atom. Consequently, three isotope values can be applied to determine the contribution of main $N_2O$ production pathways and the progress of $N_2O$ reduction. The very first attempt to jointly model $N_2O$ mixing and fractionation using Bayesian statistics was presented by Toyoda et al. [23], where two isotopic signatures $\delta^{15}N$ and SP were applied to distinguish between two mixing sources and to estimate the isotopic fractionation associated with $N_2O$ reduction. Later studies further developed this approach by including also $\delta^{18}O$ to this evaluation [24, 25]. Finally, a complete modelling approach based on all three $N_2O$ isotopic signatures was proposed and validated in a recent study by Lewicka-Szczebak et al. [21].

Here, we present a detailed description of the developed model that includes both stable isotope mixing and fractionation of compounds originating from sources characterised by different stable isotope composition: isotope FRactionation And Mixing Evaluation model (FRAME). To make this modelling approach widely applicable, we propose software with a user-friendly graphical interface. The stable isotope fractionation is added to the mixing model as an additional parameter and may be defined by the user for different requirements of a particular isotope system, e. g., fractionation in an open or closed system, equilibrium fractionation, or other special cases. The model can integrate up to three isotopic signatures of each compound. Another novel feature of this model is the introduction of isotope mixing sources as ranges of values showing equal probability instead of assuming the highest probability for the mean values. This approach is much more relevant to real-life conditions for most stable isotope case studies. FRAME was initially designed and tested for application in stable isotope studies. However, this tool can be used with many other tracers in different research areas, because the mathematical formulas are universal for any mixing cases with the possible addition of processes that alternate the characteristics of the final mixture. The primary aim of this study was to develop easy-to-use software with a flexible mathematical algorithm and comprehensively test it using several case studies.

## 2 Methods

### 2.1 Partitioning contributions from different sources to mixtures using stable isotopes

The stable isotope composition of the various elements in a chemical compound can be analysed simultaneously. Sometimes stable isotope composition can also be determined for a specific atom site position in a molecule. Therefore, several stable isotope signatures can be identified for each compound in a sample. The stable isotope compositions are usually reported using delta notation in permille (‰, as $1000 \times \delta$), e. g.:

$$\delta^{18}O = R_{\text{sample}}/R_{\text{standard}} - 1 \qquad (1)$$

where $R$ is stable isotope ratios between heavier and lighter isotope (e.g., $R = {}^{18}O/{}^{16}O$) in the analysed material and internationally recognised reference material defining the zero point of the stable isotope scale (e.g., VSMOW-SLAP). The stable isotope composition of a sample representing a mixture can be used to disentangle relative fractions from the sources contributing to the mixture using mass balance models (found by means of analytic calculation and standard propagation of uncertainties, see Sec.3.1.1, 3.2.1). The exact calculations are possible for certain cases when the number of sources is low and they have significantly different stable isotope signatures. However, three commonly encountered conditions render algebraic calculation impossible:

- The number of sources $m$ is greater than the number of stable isotope signatures plus one ($> n + 1$).

- The source stable isotope composition is defined as a non-normal probability distribution (e. g. uniform distribution within a given range).

- Additional processes are affecting the isotopic signatures, i. e. isotope fractionation or equilibration.

Solutions to this category of problems are frequently sought using Monte Carlo numerical methods. In this study, a Markov-Chain Monte Carlo (MCMC) algorithm was implemented to evaluate the mixing fractions and the contributions from fractionation processes, their correlation and uncertainties. The model is based on the Metropolis-Hastings algorithm (Sec. 2.3).

### 2.2 Input data and mixing model

The MCMC model calculates the fractional contributions from individual sources based on the measured isotopic signatures and a declared model equation that describes the mixing and other possible processes affecting the mixture. The following variables need to be identified to start the calculations:

- Sample measurements, $x = (x_1, x_2, \ldots)^T$ (vector, $T$ annotates the transposition), where $x_1$, $x_2, \ldots$ refer to measured stable isotope signatures of the sample, e. g. $\delta^{18}O$ or $\delta^{15}N$. The number of measured isotope signatures $n$ ($1 \leq n \leq 3$) determines the dimension of the modeled stable isotope system (the following short-hand notation is used throughout the article: 1D, 2D, 3D).

- Stable isotope composition of potential sources contributing to the mixture $S = (S_1, S_2, \ldots, S_m)$. Each source is represented by an $n$-dimensional vector $S_i = (S_{1,i}, S_{2,i}, \ldots)^T$, where $S_{j,i}$ is the measured mean corresponding to $i$-th considered isotope signature ($x_1, x_2, \ldots$). The number of sources $m$ does not have any upper limit.

- Optional auxiliary parameters $A = (A_1, A_2, \ldots)^T$ that describe the fractionation process that modifies the model on top of the standard mixing of sources (see Sec. 5 for examples). There is no upper limit on the number of auxiliary parameters.

Monte Carlo integration is based on the model equation given by the user, which describes the mixing process and any other processes that might influence the stable isotope composition. In the simplest scenario, when it is expected that no process other than mixing is affecting the sample composition, the model takes the following form (later referred to as $\mu_0$):

$$\mu = \mu_0 = \sum_{i=1}^{m} f_i \, S_i \qquad (2)$$

The model equation takes a different form when other processes affect the final isotope composition of a mixture, such as oxidation, reduction, evaporation, radioactive decay, etc, allowing for the integration of auxiliary parameters ($A$) and variables ($r$) to be estimated:

$$\mu = \mu_0 + \mu_{\text{aux}}(A, r) \qquad (3)$$

Examples of the inclusion of fractionation processes are discussed in detail in Sec. 3.3.

## Glossary

- $n$ – number of measured isotope signatures (e. g. $\delta$-values)

- $\mathbf{x} = (x_1, x_2, \ldots)^T$ —vector of $n$ measured isotope signatures

- $m$ —number of sources

- $S = (S_1, S_2, \ldots, S_m) = \begin{pmatrix} S_{1,1} & S_{1,2} & \ldots & S_{1,m} \\ S_{2,1} & S_{2,2} & \ldots & S_{2,m} \\ \ldots & & & \end{pmatrix}$

– matrix of measured mean characteristic isotopic signatures of the sources

- $A = (A_1, A_2, \ldots)^T$ —vector of auxiliary parameters.

- Model equation:

$$\mu = \mu_0 + \mu_{\text{aux}}(A, r), \qquad \mu_0 = \sum_{i=1}^{m} f_i \, S_i$$

$$(\text{mixing} + \text{fractionation}) \qquad (\text{mixing})$$

- Model variables:

  - sources fraction contributions:

$f_1, f_2, \ldots, f_m$, where $\sum_{i=1}^{m} f_i = 1$ and $f_i \in [0, 1]$

  - auxiliary variables: $r_1, \ldots,$ where $r_i \in [0, 1]$

## 2.3 Metropolis-Hastings algorithm

The Markov-chain Monte Carlo procedure implemented in FRAME uses the Metropolis-Hastings algorithm to scan the phase-space of possible solutions. The probability distributions of the drawn random variables were taken in the most general variants (uninformative priors) as given below in the description of the algorithm. The simulation begins with setting the threshold value $T = 0$ and then for each sample the following list of instructions is executed:

- Read vector of measurements $x$ and $\sigma(x)$ for its uncertainties (e.g. isotope delta value).

- For each iteration:

  1. Draw random variables $f_i$ from Dirichlet distribution, where $i \in [1, m]$ iterates the sources, $f = (f_1, f_2, \ldots, f_m)$.

  2. If there are any auxiliary variables: draw random variables $r_i$ from uniform distribution in $[0, 1]$ (where $i$ iterates auxiliary variables), $r = (r_1, r_2, \ldots)$.

  3. Draw random variable $\alpha$ from uniform distribution in $[0, 1]$.

  4. Calculate the model equation $\mu(f, r)$.

  5. Calculate the likelihood function $L(x|\mu)$.

  6. If $L(x|\mu) \geq \alpha T$:

     - set $T = L$,

     - if burnout is finished: append drawn variables $(f, r)$ to the Markov chain.

  7. If Markov chains achieve the desired length or if the number of iterations is exhausted, finish and store the Markov chains.

For each of the simulated variables $(f_1, f_2, \ldots, r_1, \ldots)$ a chain of accepted entries (Markov chain) is obtained from which a distribution can be calculated together with correlations of each pair of variables. For the final result, the mean of the distribution is taken and the statistical uncertainties are evaluated by finding the limits that enclose 68.2% ("$1\sigma$") of the distribution. The interpretation of the results and their uncertainties are explored in greater detail in Sec. 4.1.

## 3 FRAME in simple examples

The implementation of MCMC in the FRAME model allows for a precise determination of the range of possible solutions to isotope mixing problems and the correlations between evaluated variables. In this section, a set of simple theoretical examples is presented to explain the applied solutions and the uncertainty in the computation interpretation. To acquire experience in interpretation how the data uncertainties and the size of the model's solution phase space influence the final results, we examine examples with increasing levels of complexity. First, 1D calculations are shown and we present how data uncertainties are accounted for in both analytical and model computations. The description continues with a generalization of the problem into 2D models and we show a set of exemplary model calculations for common cases. This section concludes with a demonstration of how the process of fractionation is included in the 2D model.

### 3.1 1D mixing calculation and modeling

**3.1.1 Analytical calculations for two sources in 1D.** The simplest case study can be formulated as a mixing of two sources ($S_1$ and $S_2$). The source ($S_1$ and $S_2$) and the mixture ($x$)

isotopic signatures are known with perfect accuracy (there are no uncertainties). This scenario can be described using the following equations:

$$\begin{cases} x = f_1 S_1 + f_2 S_2 \\ f_1 + f_2 = 1 \end{cases} \tag{4}$$

and can be simply solved, obtaining:

$$f_1 = \frac{S_2 - x}{S_2 - S_1}, \qquad f_2 = \frac{x - S_1}{S_2 - S_1} \tag{5}$$

The next step is to explore how this case can be solved when finite uncertainties are considered. The uncertainties of measurements of $x$, $S_1$ and $S_2$ are given by: $\sigma_x$, $\sigma_{S_1}$ and $\sigma_{S_2}$ respectively. Using standard methods of error propagation, the following uncertainty estimation for $f_1$ is obtained:

$$\sigma_{f_1}^2 = \left| \frac{\partial f_1}{\partial x} \right|^2 \sigma_x^2 + \left| \frac{\partial f_1}{\partial S_1} \right|^2 \sigma_{S_1}^2 + \left| \frac{\partial f_1}{\partial S_2} \right|^2 \sigma_{S_2}^2 \tag{6}$$

and analogically for $f_2$. Note, that now the solution is no longer a single number, but instead it is given as a Gaussian-like probability distribution with a maximum at $f_1$ ($f_2$) and width $\sigma_{f_1}$ ($\sigma_{f_2}$). Alternatively, a case can be considered in which the source isotopic signatures do not give a particular value (with uncertainty), but instead are treated as a range of values characteristic of this source. It is then assumed that $S_i$ can take any value in the range $(S_i - \Delta S_i, S_i + \Delta S_i)$ with equal probability. The solution was modified into the following form:

$$f_1 \in \left( \frac{S_2 - \Delta S_2 - x}{S_2 - \Delta S_2 - S_1 + \Delta S_1}, \; \frac{S_2 + \Delta S_2 - x}{S_2 + \Delta S_2 - S_1 - \Delta S_1} \right) \tag{7}$$

$$f_2 \in \left( \frac{x - S_1 - \Delta S_1}{S_2 + \Delta S_2 - S_1 - \Delta S_1}, \; \frac{x - S_1 + \Delta S_1}{S_2 - \Delta S_2 - S_1 + \Delta S_1} \right) \tag{8}$$

Such a solution is no longer unique, instead, a "phase space" of possible solutions is obtained and the final result is given as an uniform probability distribution bounded by the limits given in the above equations.

**Two ways of defining source $S_i$:**

- **Point with uncertainty**: $S_i \pm \sigma_{S_i}$

In this case the source is described by a well defined mean value with analytical uncertainty, which represents the highest probability (see Eq 12).

- **Range of equally probable values**: $S_i \in (S_i - \Delta S_i, S_i + \Delta S_i)$

The natural variance of the source is described by the range of values showing equal probability for each value within this range (see Eq 13).

**Which option to use?**

In nature, most sources have a certain natural variance rather than a discrete value. Therefore, the range of equally probable values usually better describes the real-life

scenario and whenever possible the full range should be determined. Moreover, this range can be further broadened with the addition of analytical uncertainty of measured source isotope composition, which is added as a margin at the edges of the range (see the S1 Appendix for details).

The format of point-like source can be applied for precisely determined source values, e. g. in experiments conducted for the particular case studies.

**3.1.2 Modelling in 1D.** In this section, modelling of the 1D case will be studied, which, although trivial, will serve as an introduction to the methods and terminology used. The Monte Carlo model will be considered (as defined in Sec. 2.3) to obtain the solution by maximizing a likelihood function $L$. Depending on the source definition, it will be formulated as follows:

1. Source $S_i$ is defined as a point with uncertainty:

$$L(x|\mu, \sigma) = \frac{1}{\sigma\sqrt{2\pi}} \exp\left(\frac{-(x - \mu)^2}{2\sigma^2}\right) \qquad (9)$$

where $L(x|\mu, \sigma)$ is the likelihood of measurement $x$ given calculated $\mu$ and $\sigma$, which are defined as:

$$\mu = f_1 S_1 + f_2 S_2, \qquad \sigma^2 = f_1^2 \sigma_{f_1}^2 + f_2^2 \sigma_{f_2}^2$$

2. $S_i$ is defined as a finite uniform distribution (see Appendix A in S1 Appendix for details):

$$L(x|\mu, \sigma, \Delta) = \mathrm{erf}\left(\frac{x - \mu - \Delta}{\sqrt{2\sigma^2}}\right) - \mathrm{erf}\left(\frac{x - \mu + \Delta}{\sqrt{2\sigma^2}}\right) \qquad (10)$$

where again $L(x|\mu, \sigma, \Delta)$ is the likelihood of measurement $x$ given calculated $\mu$, $\sigma$ and $\Delta$ which are defined as:

$$\mu = f_1 S_1 + f_2 S_2, \qquad \sigma^2 = f_1^2 \sigma_{f_1}^2 + f_2^2 \sigma_{f_2}^2, \qquad \Delta = \frac{\partial\mu}{\partial S_1}\Delta S_1 - \frac{\partial\mu}{\partial S_2}\Delta S_2$$

Note that $\sigma_{f_1}$ and $\sigma_{f_2}$ now do not include uncertainties of $S_i$, unlike in the example above. Instead, a new variable $\Delta$ (later referred to as *spread*) accounts for their finite variability in the $(\mu - \Delta, \mu + \Delta)$ range.

In the example calculations for this case study, the following parameters were used for defining the sources and the sample (Table 1), both assuming that fractions $f_1$ and $f_2$ are equal to 0.5:

The results obtained with both variants of the model (Table 2) can be confronted with true values of $f_1$ and $f_2$ and analytically calculated uncertainties. The verification can be followed as:

$$\sigma_{f_i}^2 = \left|\frac{\partial f_i}{\partial x}\right|^2 \sigma_x^2 + \left|\frac{\partial f_i}{\partial S_1}\right|^2 \Delta S_1^2 + \left|\frac{\partial f_i}{\partial S_2}\right|^2 \Delta S_2^2 \qquad (11)$$

Parameters $\sigma_{f_i}$ are simply standard deviations of obtained distributions and the confidence intervals were calculated as described in Sec. 4.1.

**Table 1. Input values used for the calculations of 1D examples.** See the text for details.

| | example 1 | | | example 2 | | |
|---|---|---|---|---|---|---|
| | value | uncertainty | | value | uncertainty | spread |
| $x$ | 11 | 0.1 | $x$ | 11 | 0.1 | - |
| $S_1$ | 2 | 2 | $S_1$ | 2 | - | 2 |
| $S_2$ | 20 | 4 | $S_2$ | 20 | - | 4 |

**Table 2. Model outputs for the mixing case study defined in Table 1.**

| | $f_1$ | $f_2$ | $\sigma_{f_1}$ | $\sigma_{f_2}$ | $f_1$ 68% CI | $f_2$ 68% CI |
|---|---|---|---|---|---|---|
| analytical | 0.500 | 0.500 | 0.124 | 0.124 | - | - |
| model, example 1 | 0.495 | 0.506 | 0.098 | 0.098 | 0.397–0.593 | 0.408–0.604 |
| model, example 2 | 0.475 | 0.525 | 0.099 | 0.099 | 0.420–0.539 | 0.469–0.587 |

Note that the uncertainties and the spread of sources isotopic signatures are affecting the calculations through different mechanisms, thus have significantly different influences on the variables' distribution and in consequence on the results and their uncertainties (Table 2, Fig 1). In each analysis, an important choice must be made regarding which mechanism of defining sources is more suitable for a given problem. In most cases a combination of both approaches will be the most accurate, provided that enough information on the source determination is available.

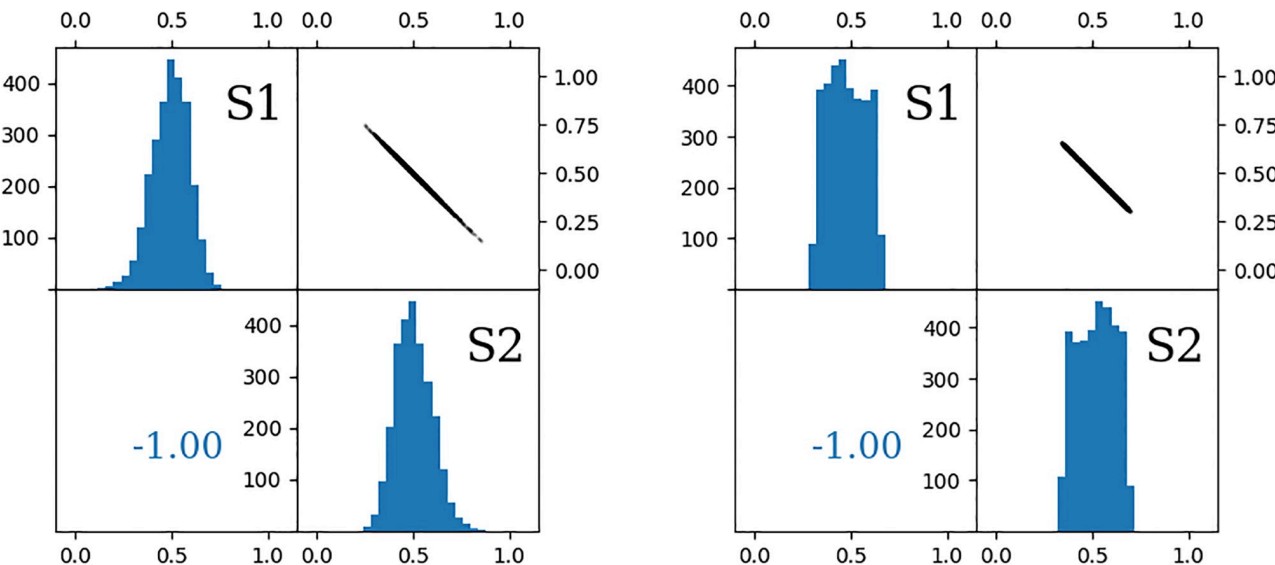

**Fig 1. Results of FRAME calculation of two simple 1D examples: One treating the sources as points with uncertainties (left) and the other assuming their flat probability distribution with a certain range (right).** Equal contribution of both sources was assumed and exact input values are listed in Table 1. The histograms on the diagonals shows distributions $f_1$ and $f_2$ built with the simulated Markov chains. The horizontal axis represents the values of $f_1$, $f_2$ fractions and the vertical axis marks the number of counts (iterations). The top right panels show the scatter-plot of all entries, where abscissa is the value of $f_1$ and ordinate shows $f_2$. The bottom left panel shows the correlation coefficient.

### 3.2 2D mixing calculation and modelling

**3.2.1 Analytical calculations in 2D.** The presented 1D calculations are now repeated for 2D case studies and they can be further generalized to higher dimensions. The simplest case with two measured isotope signatures and three sources contributing to the mixture can also be considered in two primary variants (Fig 2, Sec. 3.1):

1. sources as points with uncertainties,

2. sources as a range of possible values.

In both cases, the calculation can be described using the following set of equations:

$$\begin{cases} \boldsymbol{x} = \begin{pmatrix} x_1 \\ x_2 \end{pmatrix} = \begin{pmatrix} S_{1,1} & S_{1,2} & \cdots \\ S_{2,1} & S_{2,2} & \cdots \end{pmatrix} \begin{pmatrix} f_1 \\ f_2 \\ \cdots \end{pmatrix} \\ f_1 + f_2 + \cdots = 1 \end{cases} \tag{12}$$

Note that the above set of equations is solvable only if the number of sources does not exceed 3, otherwise there is no unique solution. Similarly, assuming a finite spread of sources' signatures, the solutions will not be unique. Still, however, the phase space of possible solutions can be studied and it may provide some limits on the $f_i$ values and correlations between fractions $f_i$. This can be done with MCMC modelling as described in the following section.

**3.2.2 Modelling in 2D and higher dimensions.** The modelling in 2D and higher dimensions does not differ from the 1D case, although the likelihood function that the algorithm evaluates in each step is modified to account for additional dimensions in the following way:

1. **Sources as points with uncertainties**

$$L(\boldsymbol{x}|\boldsymbol{\mu}, \boldsymbol{\sigma}) = \prod_{i=I_1, I_2, \cdots} \frac{1}{\sigma_i \sqrt{2\pi}} \exp\left(\frac{-(x_i - \mu_i)^2}{2\sigma_i^2}\right) \tag{13}$$

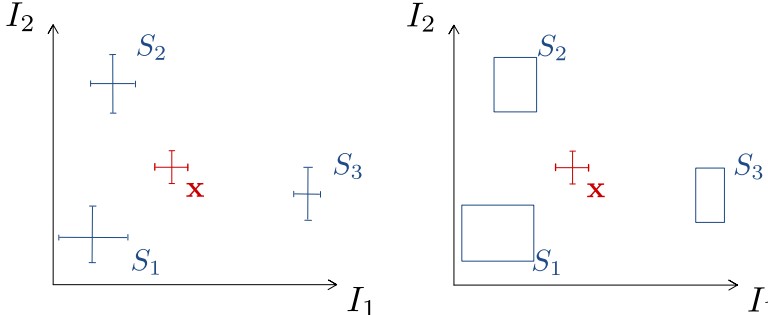

**Fig 2. Sketch of a simple 2D case study.** $I_1$ and $I_2$ span the two dimensions of the calculation and measurement $x$ originates from the mixing of three sources $S_1$, $S_2$, $S_3$. The objective was to calculate the contributions from each source. The left panel shows sources considered as points with uncertainties and the right panel shows sources as rectangles reflecting the range of possible values observed in the sources.

2. **Sources as finite ranges**

$$L(\boldsymbol{x}|\boldsymbol{\mu},\boldsymbol{\sigma}) = \prod_{i=I_1,I_2,\cdots} \left[ \mathrm{erf}\left( \frac{x_i - \mu_i - \Delta_i}{\sqrt{2\sigma_i^2}} \right) - \mathrm{erf}\left( \frac{x_i - \mu_i + \Delta_i}{\sqrt{2\sigma_i^2}} \right) \right] \tag{14}$$

where

$$\Delta_i = \frac{\partial \mu_i}{\partial S_{i,1}} \Delta S_{i,1} + \frac{\partial \mu_i}{\partial S_{i,2}} \Delta S_{i,2} + \cdots \tag{15}$$

Note that this form of model equations is valid for any number of considered sources and any number of dimensions.

**3.2.3 2D mixing with FRAME.**   This section presents a simple example of the mixing of three sources characterised by two stable isotope signatures (2D). To illustrate the basic principles of the model operation a simple hypothetical dataset was used. In this example, the contribution from each of the three sources to the mixture was exactly at 1/3. The sources were clearly separated, and all sample isotope signatures were located within the accessible domain (within a mixing polygon limited by three sources). The exact input data are listed in Table 3.

The model is initialized with a randomized configuration of $f_1$, $f_2$ and $f_3$. New configurations are drawn in each iteration and if the Metropolis condition (item *f.* in Sec. 2.3) is fulfilled it is added to the saved Markov chain. This is illustrated in Figs 3 and 4; the line-plots in the first figure show more than 800 entries accepted into the Markov chains of probed variables

**Table 3. Model input parameters for 2D calculations with two isotope signatures $I_1$, $I_2$ and three sources: $S_1$, $S_2$, $S_3$.**

| parameter | $I_1$ | $I_2$ | $\Delta_{I1}$ | $\Delta_{I2}$ | $\sigma_{I1}$ | $\sigma_{I2}$ |
|---|---|---|---|---|---|---|
| $x$ | 10 | 10 | - | - | 1 | 1 |
| $S_1$ | 0 | 0 | 1 | 1 | - | - |
| $S_2$ | 10 | 30 | 2 | 2 | - | - |
| $S_3$ | 20 | 0 | 2 | 1 | - | - |

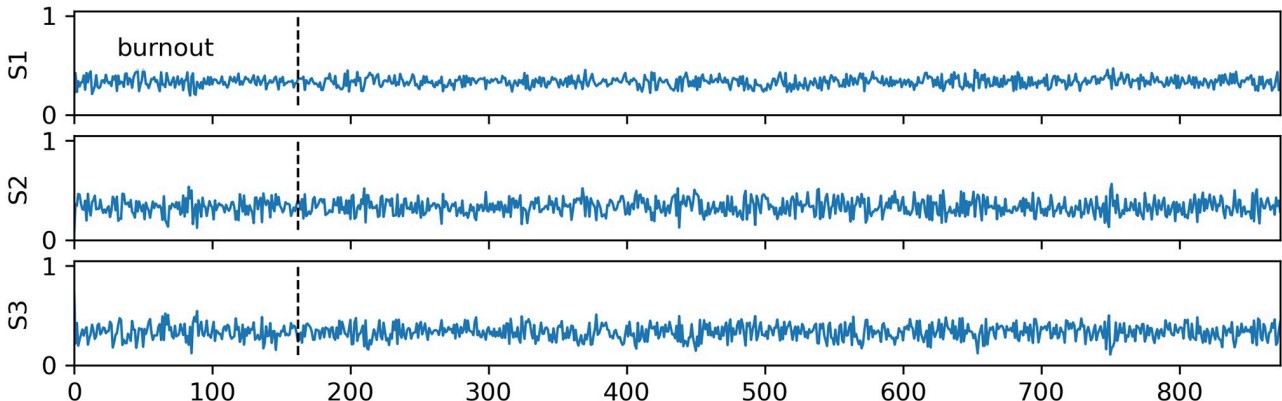

**Fig 3. Mixing configurations ($f_1$, $f_2$, $f_3$) from all model iterations that fulfilled the Metropolis condition—The Markov chain.** The number of initial iterations required for the model stabilisation is discarded (burnout, left to the dashed line), reaching the region of maximum likelihood. In MCMC simulations it is crucial that stabilization is indeed achieved, therefore such a plot is a key tool for quality assessment. In this example, the simulation begins with fractions already close to the equilibrium value, thus the stabilization is achieved virtually instantly and no visible change in oscillations is observed in dependence on the iteration number.

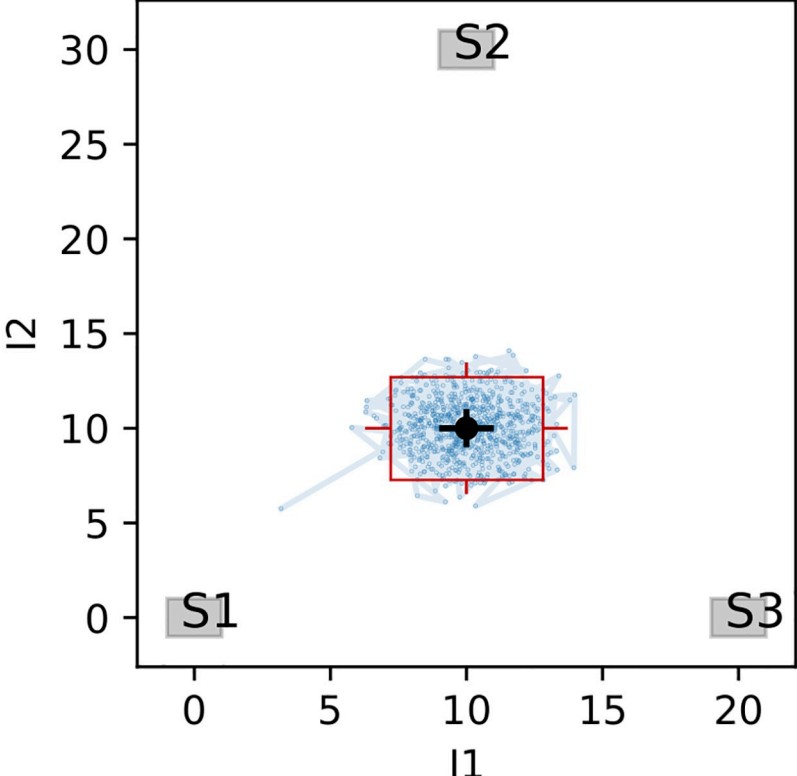

**Fig 4. The path of consecutive entries stored in the Markov chain plotted in $I_1$, $I_2$ plane.** Each dot represents a model value $\mu$ calculated for each of mixing configurations $(f_1, f_2, f_3)$ and the line connects consecutive entries. The measurement is denoted with a black dot and error bars, while the sources and their spread are represented with shaded rectangles. The red rectangle represents the boundary with the middle at $x$ and sides' length is taken as $\Delta_i$ (see Eq 15). Additional red whiskers stand for $\sigma_i$ (see Eqs 13 and 14).

representing the measured isotope signatures of the sample. The scatter-plot in the second figure also visualizes consecutive entries in the Markov chain, although the configurations are translated into $(I_1, I_2)$ space, i. e. positions in the $(I_1, I_2)$ plane. Each dot stands for an entry in the chain and consecutive dots are connected with the blue line. The portion of the initial iterations, called "burn-out" (or "burn-in"), is discarded. This initial sequence is used for the model self-stabilization, as it starts from random configurations of low-likelihood. The burn-out phase is visible in both figures as a few of the first iterations are placed far from the main cloud. No general rules indicate the necessary number of burn-out iterations, and it is not straight-forward to verify whether the stabilization was indeed achieved. In the presented examples, the number of burn-out iterations was set at an arbitrary value and then tested if it was sufficiently high. This topic is revisited in greater detail in Sec. 4.2. Both the variance in time-series and the size of the cloud of points in $(I_1, I_2)$ space reflect the data uncertainty and the spread of the sources.

The computation outcomes can be presented on the histograms (Fig 5, on the diagonal) calculated from the variables building the Markov chains along with their correlations as contour-plots (Fig 5, top-right) and correlation coefficients (Fig 5, bottom-left). The histograms show that the distribution of each of the evaluated mixing fractions has a well-defined maximum and their widths can be easily examined. Each of the estimated variable pairs shows a negative correlation, which follows expectations as all fractions should add up to unity, and

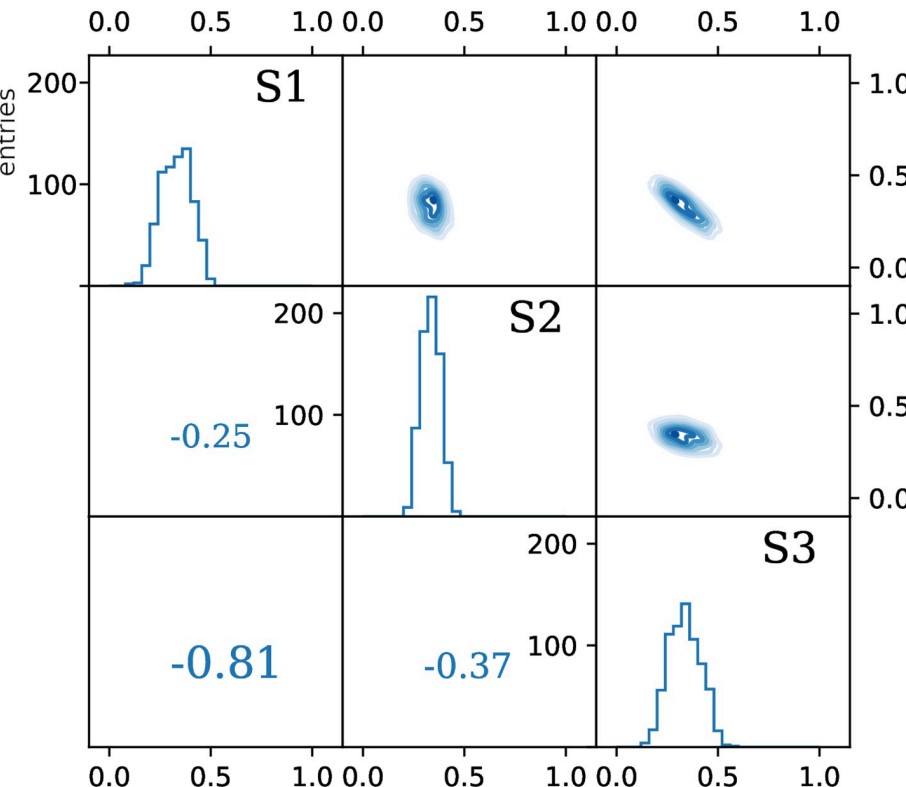

**Fig 5. Histograms (on the diagonal) calculated from the variables building the Markov chains along with their correlations as contour-plots (top-right) and correlation coefficients (bottom-left).** See the text for a more detailed analysis of the data shown.

gain in one should correspond to a loss in the other. The highest correlation is seen for the pair $S_1$, $S_3$ and also their distributions are much wider than for $S_2$. This reflects that both $S_1$ and $S_2$ have the same isotopic signature $I_2$.

Note that all calculated fractions fall very close to their true values expected from solving algebraic equations for the mixing model (Table 4). However, this alone is not sufficient to judge whether the simulation was successful and whether the model indeed describes the data. FRAME provides the set of performance plots described below, which are updated in real-time during the computation and serve as basic validation tools:

- **Markov chain time series** (Fig 3)—a purpose of this chart is to visualize the stability of calculated solutions. The desired stable output should resemble a random noise concentrated around the mean value. The uncertainty of the variable determines the spread of the noise, which frequently may be very large. The model can be considered incorrect in an unlikely event when the plot shows no stabilization or the entries are scattered around two or more

**Table 4. Results of the MCMC calculations for the mixing of the three sources.** The sources were separated very well and contributed equally to the sample mixture.

|        | mean  | median | st. dev. | CI68% low | CI68% up | true  |
|--------|-------|--------|----------|-----------|----------|-------|
| $f_1$  | 0.331 | 0.334  | 0.074    | 0.252     | 0.411    | 0.333 |
| $f_2$  | 0.334 | 0.336  | 0.047    | 0.286     | 0.383    | 0.333 |
| $f_3$  | 0.335 | 0.331  | 0.077    | 0.253     | 0.419    | 0.333 |

discerned values. In such cases, the model encounters several regions of similar likelihood and thus any further calculation will be heavily biased.

- **Optimization path** (Fig 4)—an expected output for this figure is a cloud of points, centered around the measurement, which size is determined by the uncertainties and the spread of the sources. Note, however, that each dot represents a mean model value $\mu = \mu(f_1, f_2, f_3)$, thus by construction it may never appear outside the polygon marked by the centers of the rectangles representing sources. Thus, it is possible to record a measurement outside the polygon, although within the distance not larger than the summed spread of the sources in each dimension ($\Delta_i$), plus the uncertainty ($\sigma_i$). In such cases, the cloud of simulated points is largely deformed, with a sharp edge along the polygon boundary. However, this would not mean that the model fails to describe the data and the obtained results will still be accurate. If the measurement falls beyond the discussed range, it must be assumed that either not all sources are known or there is another, unaccounted, mechanism affecting the stable isotope composition (e. g. fractionation).

- **Variable correlations** (Fig 5)—the distributions shown in this plot are drawn discarding the burnout iterations and are frequently shown as a final representation of the model results. However, the features of the drawn data may also serve as an effective sanity check. In an unlikely situation when the model finds two separated local maxima, it may also manifest in the double-peak structure of the histograms or the 2D contour plots. In such cases, the results of the simulation will be incorrect.

For a broader summary on the topic of quality assessment please consult Sec. 4.2. For other practical examples please see the S1 Appendix.

### 3.3 Auxiliary parameters expanding model calculation capacity

The unique core functionality of FRAME is its flexibility to include various auxiliary parameters that can describe processes other than mixing. The model equation can be modified to account for, e. g., the progress of stable isotope fractionation respectively to the substrate residual or reacted fractions. Any equations defined by the users can be included, but the most commonly used are the following (defined as in the glossary in Sec. 2.2):

1. **Open system fractionation**—characterized by continuous substrate replenishment and product removal. This fractionation is applicable for steady-state systems where the input and output fluxes are in equilibrium, hence the concentrations remain constant over time following the equation:

$$\mu = \mu_0 - A(1 - r) \tag{16}$$

where $\mu$ is the final stable isotope composition after fractionation, $\mu_0$ is the stable isotope composition of the initial mixture before fractionation, $A$ stands for isotope fractionation factor (reflecting the difference between product and substrate isotope composition), and $r$ for the residual unreacted fraction.

2. **Closed system (Rayleigh-type) fractionation**—characterized by the limited substrate pool and accumulated product:

$$\mu = \mu_0 + A \ln(r)$$

where $\mu$ is the final stable isotope composition after fractionation, $\mu_0$ is the stable isotope composition of the initial mixture before fractionation, $A$ stands for isotope fractionation

factor (reflecting the difference between product and substrate isotope composition), and $r$ for the residual unreacted fraction.

It is possible that such isotopc fractionation occurs after mixing, i.e. the mixture of all sources undergoes fractionation, which can be described as follows:

$$\boldsymbol{\mu} = \boldsymbol{A} \ln (r_A) + f_1 \boldsymbol{S_1} + f_2 \boldsymbol{S_2} + f_3 \boldsymbol{S_3} \tag{17}$$

or only some of the sources undergo isotope fractionation before mixing, while this fractionation do not apply for the other sources, which can be described with the following equation:

$$\boldsymbol{\mu} = f_1 (\boldsymbol{S_1} + \boldsymbol{A} \ln (r_A)) + f_2 \boldsymbol{S_2} + f_3 \boldsymbol{S_3} \tag{18}$$

Examples of these both cases have been described before for $N_2O$ studies [21].

3. **Equilibrium fractionation**—occurring due to an abiotic process of the stable isotope exchange between various phases or as a result of reversible reactions between various compounds; usually, this effect is temperature-dependent:

$$\boldsymbol{\mu} = \boldsymbol{\mu}_0 (1 - r) + \boldsymbol{E} r$$

where $\boldsymbol{\mu}$ is the final stable isotope composition after fractionation, $\boldsymbol{\mu}_0$ is the stable isotope composition of the initial mixture before fractionation, $\boldsymbol{E}$ is the stable isotope composition after complete isotopic equilibration and $r$ is the equilibrated fraction. For systems remaining in full equilibrium, $r$ equals 1 and $\boldsymbol{\mu}$ equals $\boldsymbol{E}$. Note that in some specific cases the equilibration should be considered independently for different elements of one compound characterised by different $r$. In such cases, $r$ value needs to be multiplied by an additional parameter defined separately for each isotopic signature (an example is presented in Sect. 4.2), as follows:

$$\boldsymbol{\mu} = \boldsymbol{\mu}_0 (1 - r\boldsymbol{A}) + \boldsymbol{E} r \boldsymbol{A}$$

4. Special cases of fractionation may be represented by complex equations (e. g., combing isotope fractionation and exchange), e. g., water evaporation. An example of this case is discussed in Sec.5.2.

For practical instructions on how to introduce auxiliary parameters and equations into the model see the S1 Appendix. In the example below the 2D model that includes isotope fractionation occurring in a closed system (Rayleigh-type fractionation) as per the equation from point 2 is presented. Table 5 shows the exact values used in the simulation.

Introducing the stable isotope fractionation to the model algorithm adds a degree of freedom in the simulation of sources contribution, which is pictured in Fig 6. Dashed lines project the path along which the progress of fractionation will modify the signatures of the original

Table 5. Model input parameters for 2D calculations with two isotope signatures $I_1$, $I_2$ and three sources: $S_1$, $S_2$, $S_3$.

| parameter | $I_1$ | $I_2$ | $\Delta_{I1}$ | $\Delta_{I2}$ | $\sigma_{I1}$ | $\sigma_{I2}$ |
|---|---|---|---|---|---|---|
| $x$ | 25 | 25 | - | - | 1 | 1 |
| $S_1$ | 0 | 0 | 3 | 4 | - | - |
| $S_2$ | 10 | 30 | 4 | 3 | - | - |
| $S_3$ | 20 | 0 | 3 | 4 | - | - |
| $A$ | -5 | -5 | - | - | 1 | 1 |

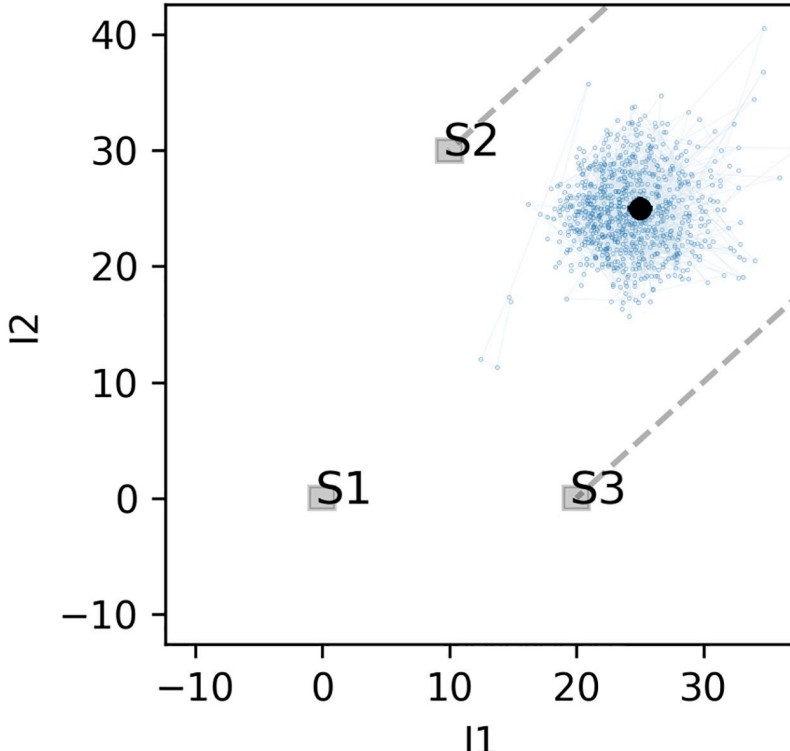

**Fig 6. Simulation of 2D mixing and fractionation—the sources $S_1$, $S_2$ and $S_3$ are pictured with grey rectangles and dashed lines show the direction and boundaries related to the fractionation process.** The measured sample is represented by a black circle and blue dots show consecutive configurations accepted by the MCMC algorithm plotted in a $I_1$-$I_2$ space.

mixture. Accounting for the fractionation causes much wider distributions of simulated outcomes and thus also larger uncertainties (Figs 7 and 8).

## 4 Reporting and validation of results

### 4.1 Results and uncertainties

The MCMC algorithm reports the results as distributions of the evaluated variables. The standard procedures of reporting the means and standard deviations are not suitable in this situation, as the distributions are confined in the range between 0 and 1 and their shapes are non-Gaussian.

In FRAME, we propose a consistent method of computing statistical uncertainties by calculating the limits, which enclose 68.28% of the total distribution around the median. This corresponds directly to the standard method of assigning $\pm 1\sigma$ (standard deviation) as the combined uncertainty while ensuring that the limits fall within the accessible domain of [0, 1] (Fig 9).

The combined uncertainty reported by FRAME accounts only for the propagation of uncertainties attributed to samples or sources as defined by the user and any systematic bias must be studied for each individual case. Possible sources of systematic uncertainties include:

- **The model is incomplete**.
  If the studied mixture is affected by any additional sources or processes that are not accounted for in the model equation, then the algorithm will not produce reliable results.

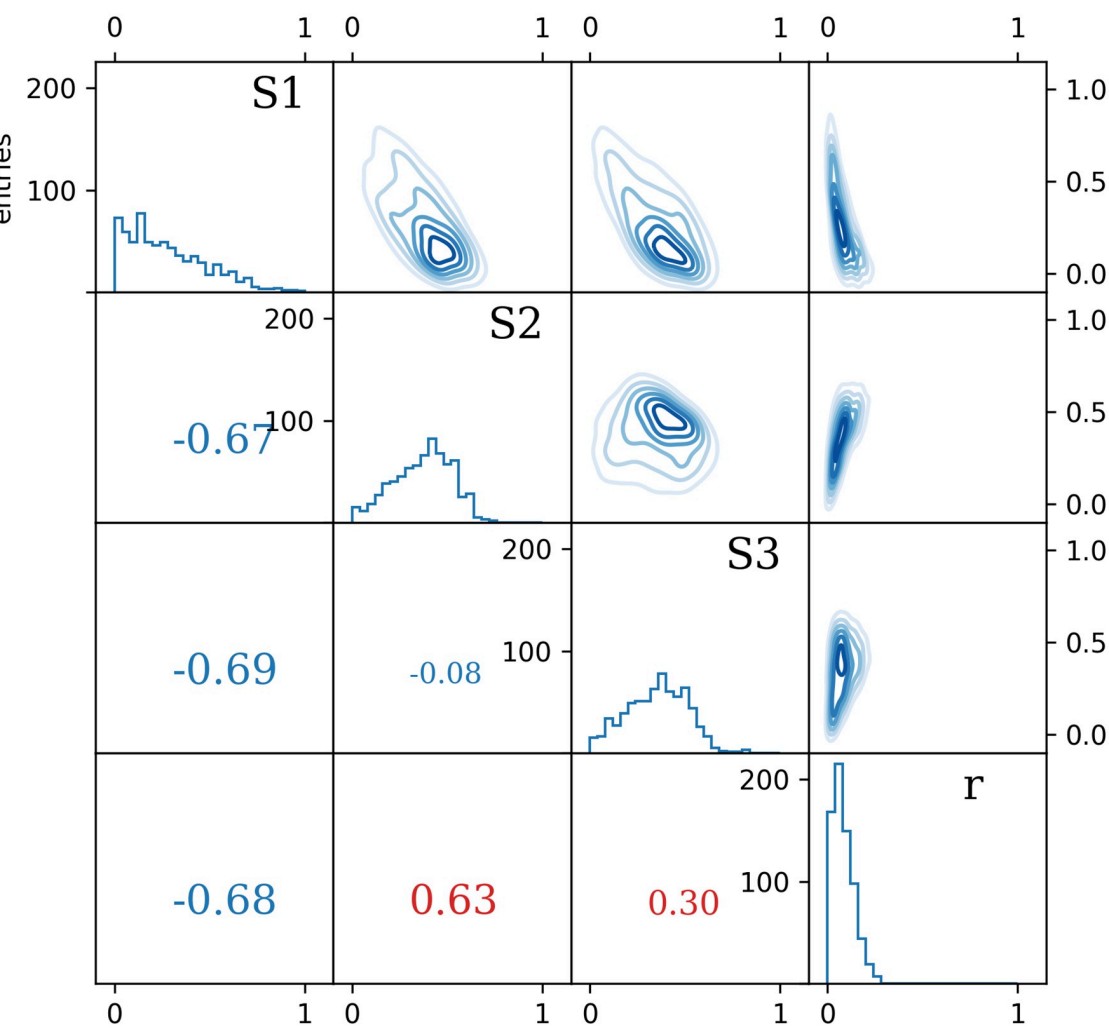

**Fig 7. The MCMC algorithm evaluated probability distributions of 3 mixing fractions: $f_{S1}$, $f_{S2}$, $f_{S3}$ and fractionation parameter $r$ describing the mixture (histograms on the diagonal) with cross-correlations (top-right) and calculated correlation coefficients (bottom-left).** Positive correlation of source $S_2$ contribution and the progress of fractionation process $r$ is due to opposite influence on the mixture (increasing $f_{S2}$ results with higher $I_1$ and $I_2$ values, while increasing $r$ does the opposite due to negative value of the parameter $A$). The non-linear shapes of the contour plots in the right-most column are due to logarithmic dependence on $r$ (see Eq 17).

Calculated fractions and fractionation factors will be biased, as the algorithm will try to compensate for a missing source or an unimplemented process.

- **The input data are flawed or inaccurate**.
  Frequently the measurements of stable isotope compositions of samples and sources are performed independently, using different methods, characterized by different uncertainties. Inconsistency in the used data sets will lead to inaccurate model output.

- **Natural variations and measurement uncertainties are not properly distinguished**.
  The distinction between natural variation and measurement uncertainties is essential for the accurate estimation of mixing. This was discussed in detail in Sec. 3.1 in the context of sources input data, although it is equally important in the case of sample measurements, in particular when the algorithm is run simultaneously for multiple data points.

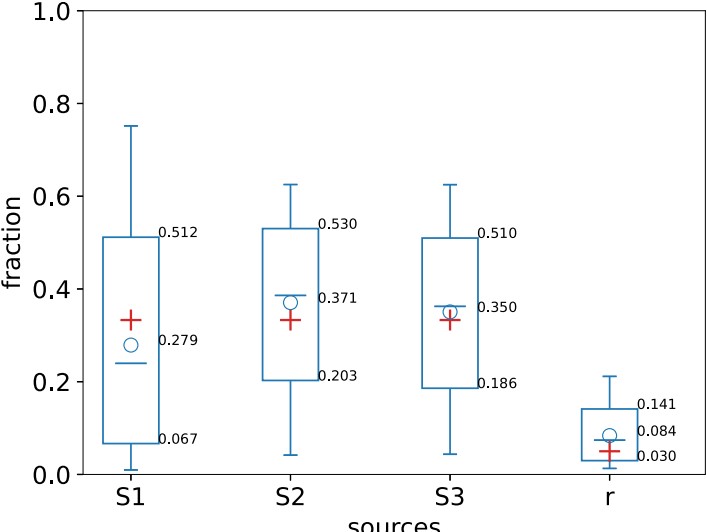

**Fig 8. Summary statistics on the simulation results: The circle stands for the mean value, the horizontal line represent the median, the box encloses 68% confidence interval (CI) and the whiskers stand for 95% CI.** The red crosses show the true values used to create the simulated dataset.

## 4.2 Quality assessment and validation

FRAME provides several tools that are useful for assessing the correctness and significance of the obtained results. The set of control plots is generated in real-time during the running of

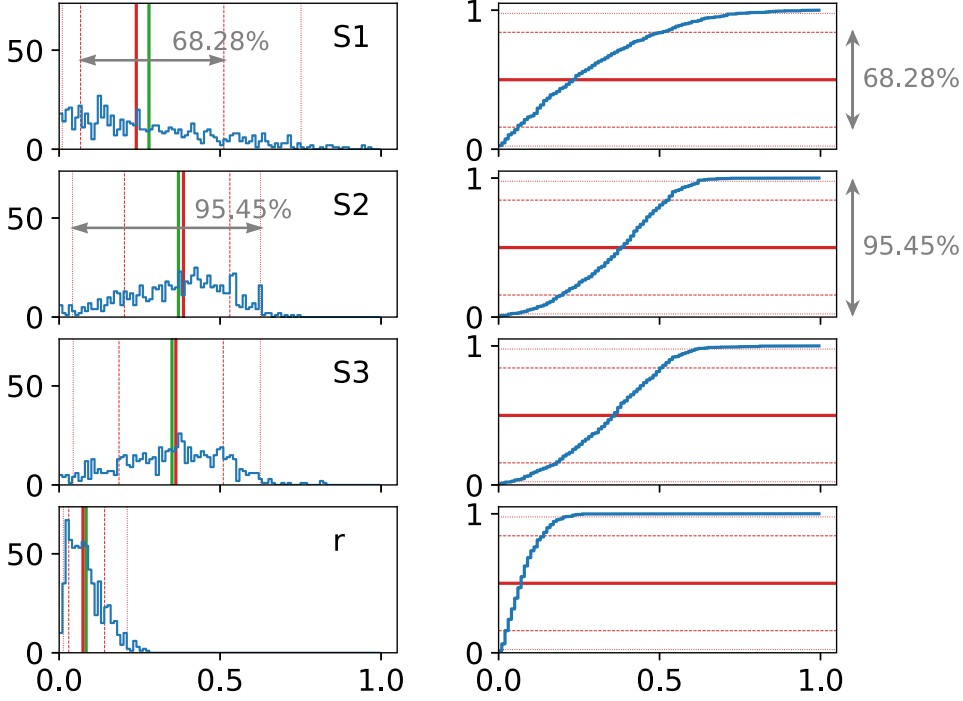

**Fig 9. Example calculation of statistical uncertainty based on the simulation described in Sec. 3.3.** The cumulants of evaluated variables are used to determine the boundaries of 68.28% ("1σ", dashed red lines) and 95.45% ("2σ", dotted red lines) confidence intervals. Solid green and red lines stand for mean and median, respectively.

the algorithm, showing basic properties and the progress of Markov chain generation (see: Sec. 3.2.3).

The first crucial step for verification of MCMC algorithm correct implementation is confirmation that the Markov chain converges to a stationary distribution. Since the calculations in the discussed examples are rather simple from a numerical perspective, the convergence is usually rapid ($\lesssim 10$ steps) and instabilities were not observed. Although, if the convergence is not apparent for a particular case study, the stability of the algorithm can be tested by running several independent simulations and comparing the distributions of simulated variables (for example with a two-sample $z$-test).

The second step is to confirm that the experimental data can indeed be described by the implemented model. While there is no definite criterion that would mark an incorrect model description, FRAME provides a statistical measure of how accurate the solution is with respect to the measured data. In the case of the Gaussian-like likelihood function this can be assessed by performing a $z$-test:

$$z = \frac{|x - \bar{\mu}|}{\sigma/\sqrt{n}} \tag{19}$$

where $x$ is the stable isotopic signature, $\bar{\mu}$ is an averaged value calculated from Markov chain entries, $\sigma$ stands for its spread and $n$ is the number of entries. However, in a situation in which the finite natural variation in sources (spread) is considered a slightly modified approach needs to be used. The $z$ score will be set at zero when the average model value $\bar{\mu}$ lies within the bounds given by $\pm\Delta$ around the measured sample and if it lies beyond, the previous equation will be modified in the following way:

$$z = \frac{\min(|x - \bar{\mu} - \Delta|, |x - \bar{\mu} + \Delta|)}{\sigma/\sqrt{n}} \tag{20}$$

Fig 10 illustrates both described cases (Eqs 19 and 20). The value of the $z$-score can be converted using standard methods to the probability of rejecting the hypothesis that the given data was measured under the assumed model ($1 - P(\boldsymbol{x}|\text{model})$), i. e. a $z$-score of 2 corresponds to a probability of $\approx 95\%$, $z$ score of 3 to $\approx 99.7\%$, etc.

Once the fulfilment of the described conditions is assured it is safe to explore and interpret the distributions of evaluated variables.

## 5 Case studies of FRAME applications

### 5.1 2D model for determination of nitrate sources and fractionation

An example of nitrates ($NO_3^-$) source partitioning is presented to illustrate the 2D stable isotope model based on the stable oxygen and nitrogen isotope composition ($\delta^{18}O_{NO_3}$ and

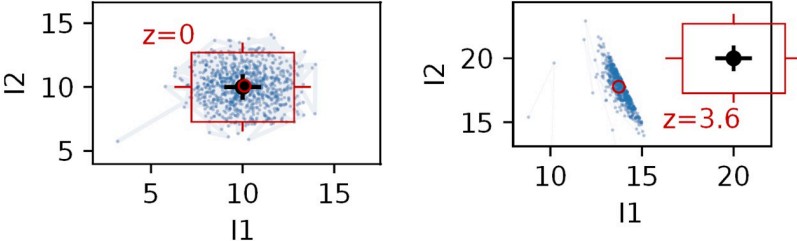

**Fig 10.** Left panel: The $z$ score equals zero when the average model value $\bar{\mu}$ lies within the bounds given by $\pm\Delta$ around the measured point. Right panel: Sample data is well beyond the bounds and the $z$-score calculated for such configuration corresponds to less than 0.01% probability of measured data under the assumed model.

$\delta^{15}N_{NO_3}$). These isotopic signatures are frequently used to trace nitrate sources in groundwaters [11–13] or surface waters [14–16] thanks to differences in various nitrate sources. However, nitrates may also undergo intensive denitrification processes, which can significantly alter their final isotopic signatures [17–19]. Therefore here we consider two cases: (i) simple mixing neglecting possible further denitrification and (ii) mixing of sources taking into account possible further isotope fractionation associated with denitrification. The new feature introduced by FRAME is the possibility of estimation of progress of the denitrification process (as presented below in case 2).

This is a fictive example of surface waters impacted by different nitrate sources. A river flowing from a pristine forested area located in the mountains carries moderate loads of nitrates (1.0 mg/L) originating from the biological activity and organic matter decomposition in the catchment. The stable nitrogen and oxygen isotope compositions of nitrate analyzed at sampling point A ($\delta^{15}N_{NO_3}$ 0.5 ± 0.5 ‰, $\delta^{18}O_{NO_3}$ 2.5 ± 0.5 ‰) primarily reflect the mean $\delta^{15}N_{ORG}$ of organic nitrogen in the catchment modified by fractionation processes leading to nitrate generation. The $\delta^{18}O_{NO_3}$ value reflects the typical contribution from water ($\delta^{18}O_{H_2O}$ -8.0 ‰) and atmospheric oxygen ($\delta^{18}O_{O_2}$ 23.5 ‰) during nitrification (ratio 2:1 [19]). After passing through the agriculture area (sampling point B), the nitrate concentration in the river water increases to 5 mg/L, and its stable nitrogen and oxygen compositions change to $\delta^{15}N_{NO_3}$ -8.3 ± 0.5 ‰ and $\delta^{18}O_{NO_3}$ 8.1 ± 0.5 ‰. It is implied that a 4 mg/L increase in $NO_3$ concentration can be attributed to agro-pollutants leaching from the surrounding field: 1) overused fertilizer, $(NH_4)_2SO_4$, leaching from crop fields, and 2) manure leaching from the cattle pasture. The applied fertilizer stable isotope composition was analyzed as $\delta^{15}N_{NH_4}$ 0.5 ‰ and the mean soil moisture $\delta^{18}O_{H_2O}$ was estimated to be 5.0 ± 5.0 ‰. Manure $\delta^{15}N_{NH_4}$ analyzed in trenches on the pasture was determined as 20.0 ± 3.0 ‰ and water $\delta^{18}O_{H_2O}$ as -5.0 ± 0.5 ‰. The stable nitrogen isotope fractionation during the nitrification ($NH_4$–$NO_3$) results in -17.0 ± 6.5 ‰ decrease in $\delta^{15}N$ during nitrification of $NH_4$ to $NO_3$ [17]. Contribution to nitrate oxygen during $NH_4$ nitrification was considered as 2:1 [19] between available water oxygen and atmospheric oxygen (23.5 ‰ [26]), Eq 17.

$$\delta^{18}O_{NO_3} = \frac{1}{3}\delta^{18}O_{H_2O} + \frac{2}{3}\delta^{18}O_{O_2} \tag{21}$$

**Case 1.** This simplified case assumes complete transformation of unused by vegetation ammonia from fertilizer and manure to nitrate and no fractionation associated with nitrates denitrification in the river or groundwaters. In the downstream river at point B, we assume based on algebraic calculations the stable isotope signatures representing the following contributions: 20% of natural $NO_3^-$ derived with the upstream river ($f_R$), 60% of fertilizer $NO_3^-$ ($f_F$), 20% of manure $NO_3^-$ ($f_M$). The FRAME model estimated the contribution to nitrate at point B as follows (Table 7): $f_R$ = 19.5 ± 11.2%, $f_F$ = 59.4 ± 5.4%, and $f_M$ = 21.2 ± 12.8%.

**Case 2.** This case also considers the possible denitrification of nitrates before the homogeneous mixture of nitrates from all sources arrives at sampling point C (downstream river). However, this should be considered approximate only because the leaching of agro-pollutants occurs over an extended time and is always transitional. The constant input of a new substrate is accompanied by constant removal of the denitrification gaseous products—$N_2O$ and $N_2$ in an open system. Therefore, this scenario reflects a proportional mixing at the time of sampling respectively to nitrates retention time and denitrification pace:

$$\boldsymbol{\mu} = f_R\,\boldsymbol{S}_R + f_F\,\boldsymbol{S}_F + f_M\,\boldsymbol{S}_M - \boldsymbol{E}\,r \tag{22}$$

**Table 6. Summary of input values for the FRAME model.** Source recalculated signatures reflect fractionation during $NO_3$ generation. $\delta^{18}O$ is the stable isotope composition of water in the place of $NO_3$ formation surface water in trenches contaminated by manure (-5.0 ± 0.5 ‰) and soil moisture (5.0 ± 0.5 ‰).

| | Measured | | | | Sources recalculated | |
|---|---|---|---|---|---|---|
| | $\delta^{15}N_{NH_4}$ | $\delta^{15}N_{NO_3}$ | $\delta^{18}O_{NO_3}$ | $\boldsymbol{\delta^{18}O_{H2O}}$ | $\delta^{15}N_{NO_3}$ | $\delta^{18}O_{NO_3}$ |
| | [‰] | [‰] | [‰] | [‰] | [‰] | [‰] |
| $x_1$: River, point B | | -8.3±0.5 | 8.1±0.5 | | | |
| $x_2$: River, point C | | 1.2±0.5 | 12.9±0.5 | | | |
| $S_1$: River, point A | | 5.0±0.5 | 2.5±0.5 | | 5.0±0.5 | 2.5±0.5 |
| $S_2$: Fertilizer | 0.50±0.5 | | | 5.0±5.0 | -16.5±0.5 | 11.2±0.5 |
| $S_3$: Manure | -20.0±3.0 | | | -5.0±0.5 | 3.0±3.0 | 4.5±0.5 |

where $\mu$ is the final stable isotope composition as $\delta$ value, $f_i$ is the respective fraction of particular $NO_3^-$ source and $S_i$ is the characteristic isotopic signature of this source, $E$ is the isotopic fractionation factor associated with denitrification: $\epsilon^{15}N = -15.9 \pm 2.0$ ‰ and $\epsilon^{18}O = -8.0 \pm 2.0$ ‰ [27] and $r$ is the nitrate reduced fraction—to be calculated by the model.

If applied to the previous example (Sample 1, River, point B—Table 6, $x_1$) the model output indicates a small but possible effect of denitrification with a reduced nitrate fraction of 12.5 ± 9.2%. However, the contributions of mixing sources are similar, with the clear dominance of fertilizer contribution to the nitrate present in the river at point B: $f_R$ = 14.8 ± 9.4%, $f_F$67.1 ± 9.7%, and $f_M$18.1± 11.9%. This indicates that the impact of denitrification on the stable isotope composition of Sample 1 is possible. To illustrate this process more clearly we assume Sample 2 (River, point C, Table 6, $x_2$) where 60% of the nitrate of Sample 1 is consumed due to denitrification ($r$ = 0.6). This will give the stable nitrogen and oxygen compositions of $\delta^{15}N_{NO_3}$ 1.2 ± 0.5 ‰ and $\delta^{18}O_{NO_3}$ 12.9 ± 0.5 ‰, i.e., enriched in heavy isotopes due to preferential consumption of molecules with light isotopes in the microbial denitrification process. The model estimates the contribution of particular fractions for: $f_R$ = 19.2 ± 13.0%, $f_F$ = 56.0 ± 11.5%, and $f_M$ = 24.9 ± 15.2%, and the progress of the denitirification process: $r$ = 56.7 ± 13.7%. The ranges of possible values are broadened due to the addition of 4th unknown value of $r$, but the results still met the expected values very well.

## 5.2 2D evaporation model for waters

Evaporative loss in surface waters can be determined based on $\delta^2H$ and $\delta^{18}O$ of water using the well-established method known as the Craig-Gordon model [28, 29], which allows determination of evaporation to inflow ratio ($E/I$) assuming steady-state model. This method bases on the typical evaporative slope between O and H isotopic signatures which differs from the slope characteristic for atmospheric precipitation [28]. Here we present how to incorporate the Craig-Gordon model into the FRAME tool. This has the advantage of taking both hydrogen and oxygen water isotopes simultaneously into account giving one $E/I$ value, instead of separate $\delta^2H$ based and $\delta^{18}O$ based results, which can be obtained with the existing *Hydrocalculator* tool [29]. Moreover, the uncertainties associated with sample measurements, source water determination and meteorological data can be taken into account and we can obtain information on the overall uncertainty of the $E/I$ estimation and determine the confidence interval of the possible results and their probability distribution.

To be able to use FRAME for this aim the typical equation used for $E/I$ determination:

$$E/I = \frac{(\delta_L - \delta_P)}{(\delta^* - \delta_L)m} \tag{23}$$

must be modified to a form of fractionation equation required by this model, i.e., describing the final stable isotope composition of a sample:

$$\boldsymbol{\mu} = \frac{r\boldsymbol{A}\boldsymbol{D} + \boldsymbol{\mu}_0}{r\boldsymbol{A} + 1} \tag{24}$$

where $\boldsymbol{\mu}$ stands for $\delta_L$—the final stable isotope composition of partially evaporated outflowing water from a lake, $\boldsymbol{\mu_0}$ stands for $\delta_P$—the initial stable isotope composition of inflowing water, $r$ is the evaporated fraction ($E/I$) to be calculated by the model, $\boldsymbol{A}$ is the parameter $m$ dependent on the relative humidity and ambient temperature, $\boldsymbol{D}$ is the limiting isotopic enrichment due to evaporation dependent on relative humidity, ambient temperature and stable isotope signature of ambient atmospheric vapors, which can be measured or calculated from the stable isotope composition of precipitation. The parameters $\boldsymbol{A}$ and $\boldsymbol{D}$ can be easily determined with the *Hydrocalculator* [29] or calculated analytically based on the known equations for the water balance adequate for the particular case [28, 29].

As an example, previously published data [30] were used. Taking into account the stable isotope composition of inflowing and outflowing to the lake water (Table 7) and taking into account precipitation stable isotope composition ($\delta^2H = -57 \pm 3$ ‰, $\delta^{18}O = -7.8 \pm 0.2$ ‰), known water temperature (13.2˚C) and relative humidity (0.65), the evaporation over inflow $E/I$ ratio of 0.22 was determined [30]. The above data can be also input to the *Hydrocalculator* tool [29], that calculates $E/I$ of 0.2248 based on $\delta^2H$ and 0.2206 based on $\delta^{18}O$. To calculate the evaporation loss with FRAME, in addition to the above data we also need to input: the analytical uncertainty in the determination of inflowing waters isotope signatures. The algorithm is defined as a single source model and the fractionation equation (Eq 22) is the only driver of the observed change in the outflowing water composition. The data set needs to be supplemented with the parameters $\boldsymbol{A}$ and $\boldsymbol{D}$ which are calculated based on the water temperature and relative humidity using *Hydrocalculator* or adequate equations for the water balance [28, 29] (Table 7). The parameters $\boldsymbol{A}$ and $\boldsymbol{D}$ are given with the assessed uncertainty due to the meteorological data measurement precision. The calculated mean $E/I$ determined by FRAME of 0.2232 is in excellent agreement with *Hydrocalculator* data. Moreover, FRAME output provides the 68% confidence interval for the $E/I$ from 0.1892 to 0.2616, based on the given input data uncertainty.

According to our knowledge, FRAME is the first tool using the Bayesian approach to calculate the evaporative losses integrating both water isotopes at the same time and providing the uncertainty range of the result based on all uncertainties of the input data. This simple case presented here can be expanded with the mixing cases of more water sources.

## 5.3 2D model for determination of nitrite pathways

FRAME was also tested using a laboratory case study partitioning nitrite ($NO_2^-$) sources with a 2D isotope model based on stable nitrogen and oxygen isotope compositions for a laboratory

**Table 7. Summary of input values for the FRAME model: Lakewater sample, inflowing water (single-source) and auxiliary parameters calculated from the meteorological conditions according to Craig-Gordon model.**

|  | $\delta^{18}O_{H2O}$ | $\delta^2H_{H2O}$ |
|---|---|---|
|  | [‰] | [‰] |
| $x_1$: Lakewater | −2.0 ± 0.2 | -25 ± 3 |
| $S_1$: Inflow | −5.0 ± 0.2 | -34 ± 3 |
| $A$ | 1.7881 ± 0.005 | 1.5824 ± 0.03 |
| $D$ | 5.6063 ± 0.005 | 0.2946 ± 0.03 |

experimental study. Nitrite is a central compound in N soil cycle and its isotopic signature depends on the mixture of three formation pathways: nitrate reduction (NAR), ammonium oxidation (AOX) and organic nitrogen oxidation (ORG) as well as isotopic fractionation during $NO_2^-$ consumption due to nitrite reduction (NIR) and nitrite oxidation (NIOX) [31]. Moreover, $\delta^{18}O_{NO_2^-}$ can be modified by O isotope exchange with ambient soil water [32]. The $NO_2^-$ isotope model was only recently proposed for differentiating nitrite sources in soil [33], where the natural abundance isotope model was calculated by applying a simple fitting procedure (by Excel Solver) without giving the probability ranges for the model outputs. Here, we show how the application of the FRAME modelling tool may increase the insight into the determined processes.

We show the calculations based on the mean $NO_2^-$ isotopic signatures from the laboratory experiment L1 (as published in [33], the isotopic signatures for the sources and stable isotopic fractionation factors for the sinks are adopted after the literature [33]. Input values are summarised in Table 8).

For nitrite sinks two fractionation processes associated with $NO_2^-$ reduction and oxidation need to be considered, which show opposite fractionation factors, with inverse fractionation for $NO_2^-$ oxidation [32]. For these two fractionation processes some pre-assumptions are needed to avoid model ambiguity: 1) the possible coexistence of both fractionation processes with the known NIR:NIOX ratio of 0.7 to 0.3 (as determined in the parallel $^{15}N$ tracing experiment [33]); 2) equilibrium isotopic fractionation of nitrite oxygen isotopes must be taken into account; 3) the amount of exchanged for $\delta_{18O}$ is known for this particular experiment (0.25) [33] and 4) the equilibrated nitrite value was determined from water isotope values and temperature [31] and equals +8.6 ‰. $NO_2^-$ pool in the soil is very small and dynamic. $NO_2$ usually does not accumulate, indicating that the input and sink fluxes are in equilibrium and the fractionation system should be considered as open. Hence, the nitrite model can be described using the following equation:

$$\boldsymbol{\mu} = (f_{NAR}\boldsymbol{S}_{NAR} + f_{AOX}\boldsymbol{S}_{AOX} + f_{ORG}\boldsymbol{S}_{ORG} - 0.7\boldsymbol{A} - 0.3\boldsymbol{B})(1 - \boldsymbol{C}) + 8.6\,\boldsymbol{C} \qquad (25)$$

where $f_i$ is the respective fraction of pathway contributing to the $NO_2^-$ production and $S_i$ is the characteristic isotope composition for this pathway, $A$ is the stable isotopic fractionation factor associated with $NO_2^-$ reduction to NO, $B$ is the stable isotopic fractionation factor associated with $NO_2^-$ oxidation to $NO_3^-$, $C$ is the fraction of $NO_2^-$ equilibrated with stable oxygen isotope composition of soil water (showing the final $\delta^{18}O$ value of 8.6 ‰ after complete equilibration with water). $\mu$ stands for the final $NO_2^-$ stable isotope composition after mixing of all sources

**Table 8. Summary of input values for the FRAME model: Nitrite sample (x), 3 sources (S given as ranges from/to), parameters representing isotope fractionation factors for nitrite reduction and oxidation (A, B) and extent of equilibrium $\delta^{18}O$ (C) given as mean with standard deviation.**

|  | $\delta^{18}O_{NO_2^-}$ | $\delta^{15}N_{NO_2^-}$ |
|---|---|---|
|  | [‰] | [‰] |
| $x_1$ | 11.8 ± 2.5 | 3.2 ± 1.9 |
| $S_{NAR}$ | 3.3 to 5.3 | -14.3 to -12.3 |
| $S_{AOX}$ | 16.4 to 20.4 | 65.9 to 71.9 |
| $S_{ORG}$ | 16.4 to 20.4 | 4.4 to 6.4 |
| $A$ | -4.0 ± 2.0 | -10.0 ± 2.0 |
| $B$ | 5.0 ± 2.0 | 13.0 ± 2.0 |
| $C$ | 0.25 ± 0.01 | 0.00 ± 0.00 |

**Table 9. Comparison of the results of the natural abundance nitrite model (*FRAME*) and the reference values based on $^{15}$N traced studies (*Ntrace*).** Mean values with standards deviation ($1\sigma$) of sources contribution are given.

| model | $f_{NAR}$ | $f_{AOX}$ | $f_{ORG}$ |
|-------|-----------|-----------|-----------|
| *FRAME* | 0.54 ± 0.17 | 0.08 ± 0.05 | 0.37 ± 0.21 |
| *Ntrace* | 0.53 ± 0.20 | 0.08 ± 0.03 | 0.39 ± 0.14 |

and partial $NO_2^-$ reduction or oxidation and equilibration for stable oxygen isotope compositions.

The experimental values used for the presented modelling cases were obtained in parallel experiments using the $^{15}N$ tracing approach, which can be used as a reference dataset. From the $^{15}N$ traced study estimations of the calculated fractions were obtained independently [33]. A comparison of both approaches is presented in Table 9.

The FRAME modelling results show very good agreement with the reference data (Table 9). However, applying the FRAME model also allows estimation of the overall uncertainties of these calculations and we can indicate quite wide probability ranges of the possible results. The FRAME results indicate that possibly the NAR and ORG fraction are not very precisely separated, since we observe quite wide ranges for these fractions. Consequently, the contribution of ORG, which is quite a novel finding for soil N cycle indicated by $^{15}N$ traced studies [34], may be similar to NAR. The FRAME model results agree with Ntrace outcomes and unambiguously confirms very low AOX contribution, up to $\approx 0.10$.

## 5.4 3D model for $N_2O$ source partitioning and quantification of $N_2O$ reduction

$N_2O$ data were used as an example for applying a 3D model in FRAME, since for this unique molecule we can measure three isotope signatures ($\delta^{18}O$, $\delta^{15}N$ and *SP*—'site preference' indicating the difference in $\delta^{15}N$ between the central and peripheral position of the linear $N_2O$ molecule). The final $N_2O$ isotope siganture depends both on the mixing of $N_2O$ production pathways and $N_2O$ fractionation associated with $N_2O$ reduction to $N_2$ [35]. The possible sources are the four main $N_2O$ production pathways: bacterial denitrification (bD), nitrifier denitrification (nD), fungal denitrification (fD) and nitrification (Ni). The isotope signatures for each source were determined in previous pure culture studies [22, 36] and showed quite wide ranges. $N_2O$ also undergoes significant stable isotope fractionation due to its possible partial reduction to $N_2$, which is associated with preferential breakage of the light isotope O-N bonds. As shown in previous studies this fractionation can be best described by the closed system dynamics with Rayleigh-type equation [24], as following:

$$\boldsymbol{\mu} = f_{bD}\boldsymbol{S}_{bD} + f_{nD}\boldsymbol{S}_{nD} + f_{fD}\boldsymbol{S}_{fD} + f_{Ni}\boldsymbol{S}_{Ni} + \boldsymbol{A}\ln(r) \tag{26}$$

where $f$ is the respective fraction of pathway contributing to the $N_2O$ production and $\boldsymbol{S}_i$ is the characteristic isotopic signature of this pathway, $r$ is the residual unreduced $N_2O$ fraction and $\boldsymbol{A}$ is the stable isotope fractionation factor associated with $N_2O$ reduction. $\boldsymbol{\mu}$ stands for the final $N_2O$ isotopic signature after mixing of all sources and partial $N_2O$ reduction.

Here we show the application of FRAME on the example of interpretation of experimental and field $N_2O$ stable isotope data [21]. For the model we define the ranges of each pathway based on the existing literature taking into account the mean ± one standard deviation of all literature reported values for each pathway (as outlined in [22]) and assuming equal probability for the values within this range. Additionally, the typical analytical uncertainties are added as standard deviation (as outlined in Sec. 3.2.2 and Appendix B in S1 Appendix). The

**Table 10. Summary of input data for the FRAME model: 3 samples ($x_1$, $x_2$, $x_3$) of different sources and reduction contribution, 4 sources (given as range from/to) and auxiliary parameter representing the reduction fractionation factor.**

|          | $f_{bD}$ | $f_{nD}$ | $f_{fD}$ | $f_{Ni}$ | $r$ | $\delta^{18}O_{N2O}$ [‰] | $\delta^{15}N_{N2O}$ [‰] | $SP_{N2O}$ [‰] |
|----------|------|------|------|------|-----|------------------|------------------|--------------|
| $x_1$    | 0.9  | 0.0  | 0.1  | 0.0  | 0.2 | 47.6 ± 0.5       | -35.2 ± 1.0      | 11.1 ± 1.0   |
| $x_2$    | 0.9  | 0.0  | 0.1  | 0.0  | 0.8 | 26.3 ± 0.5       | -45.0 ± 1.0      | 3.0 ± 1.0    |
| $x_3$    | 0.1  | 0.0  | 0.0  | 0.9  | 0.9 | 24.8 ± 0.5       | -54.0 ± 1.0      | 31.7 ± 1.0   |
| $S_{bD}$ |      |      |      |      |     | 17.8/20.6        | -52.8/-42.4      | -5.2/0.4     |
| $S_{nD}$ |      |      |      |      |     | 14.4/17.4        | -60.7/-53.1      | -7.0/2.8     |
| $S_{fD}$ |      |      |      |      |     | 43.4/51.0        | -44.4/-31.2      | 31.0/36.0    |
| $S_{Ni}$ |      |      |      |      |     | 21.4/25.6        | -63.9/-49.3      | 32.0/36.0    |
| $A$      |      |      |      |      |     | -15.4 ± 4.7      | -7.1 ± 2.1       | -5.9 ± 1.4   |

modelling results of $N_2O$ case study were previously validated with a parallel experimental approach of $^{15}N$ enrichment studies [21]. However, this validation only concerned the calculated $r$ values and some summarised $f$ values and so far there is no reference method able to validate all the fractions at the same time, which could be theoretically separated by the stable isotope modelling. Therefore, here the model performance was assessed for theoretical $N_2O$ measurements, assuming particular contribution of mixing pathways and reduction progress, according to the Table 10. The first two samples were defined according to the most typical pathways contribution with a clear dominance of bacterial denitrification ($x_1$, $x_2$) and differing in residual $N_2O$ fraction (from 0.2 to 0.8). Sample 3 ($x_3$) represents rather extreme case with strong dominance of the nitrification process and very low $N_2O$ reduction, which can be observed for very dry conditions.

The values of these samples were calculated with the mean stable isotope composition for respective sources and mean isotopic fractionation factors according to the values summarised most recently [22] (Table 10) without any substrates corrections, i.e. the isotopic signatures of substrates (i. e., nitrate, ammonium and soil water) are assumed 0 ‰.

From the graphical presentation of model results (Fig 11) can be seen that the separation of bD and nD fraction is not very precise, since we have a powerful correlation between these two fractions and very wide probability distributions. Both processes are characterised by very similar isotopic signatures, especially for $SP$ and $\delta^{18}O$, since this conclusion is well justified. Therefore, in the presented comparison with true values (as defined in Table 10) the bD and nD fractions are shown as a sum of these both fractions (Fig 12).

This study presents a complex case of 3D modelling that includes four possible sources and one fractionation process. In the presented example, we can see that the model can estimate the fractions contributions and $N_2O$ residual fraction within the maximum difference of 0.15. The trends and dominance of particular pathways are clearly displayed. However, the results must be carefully interpreted, taking into account the possible weak separation of fractions characterized by similar stable isotope signatures. As noted above, the bD and nD fractions cannot be precisely separated due to very similar ranges in isotopic signatures for $SP$ and $\delta^{18}O$. In such a case it is more justified to show a sum of these fractions and such a sum shows good agreement with the set values (Fig 12). The most challenging for separation is the very high fraction of Ni, as set for $x_3$, which tend to be underestimated by the model with simultaneous underestimation of the residual $N_2O$ fraction ($r$) and overestimation of the fD fraction (Fig 12). This is due to similar isotopic signatures for Ni and fD isotopic signatures (namely, the same range for $SP$ values). Nevertheless, even in this extreme example the Ni fraction is determined within the difference of 0.15 between set and mean modelled value (0.9 vs. 0.76).

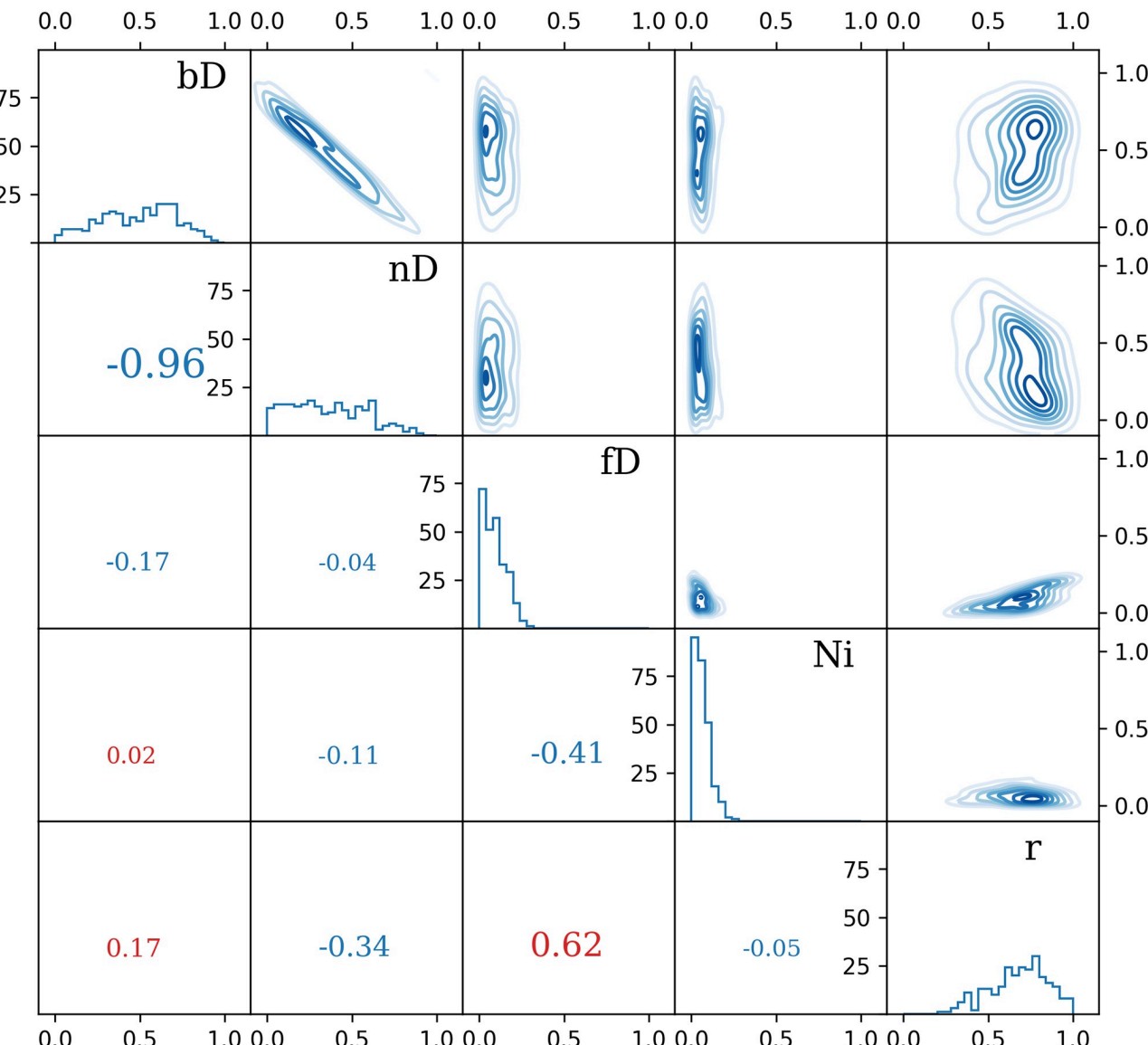

**Fig 11. Distributions and correlations of random variables $f_i$ and $r$ that constitute the Markov chain.** The histograms present the probability distribution for mixing fractions ($f$) and residual fraction ($r$), right side graphs show the correlations of respective fractions and left side values indicate $R$ values for these correlations. Sample 2 is presented.

## 6 Summary and outlook

FRAME is the first software to apply Markov-Chain Monte Carlo Modelling for partitioning sources and fractionation processes contributing to stable isotope composition of samples representing mixtures. As an output, a probability distribution of possible results is computed, which allows the assessment of the input data quality and the probability of the correct conclusions. The current model release with the additional materials and examples is available at malewick.github.io/frame.

The FRAME software was originally developed and described for stable isotope studies and any isotope mixing processes or associated stable isotope fractionations can be implemented

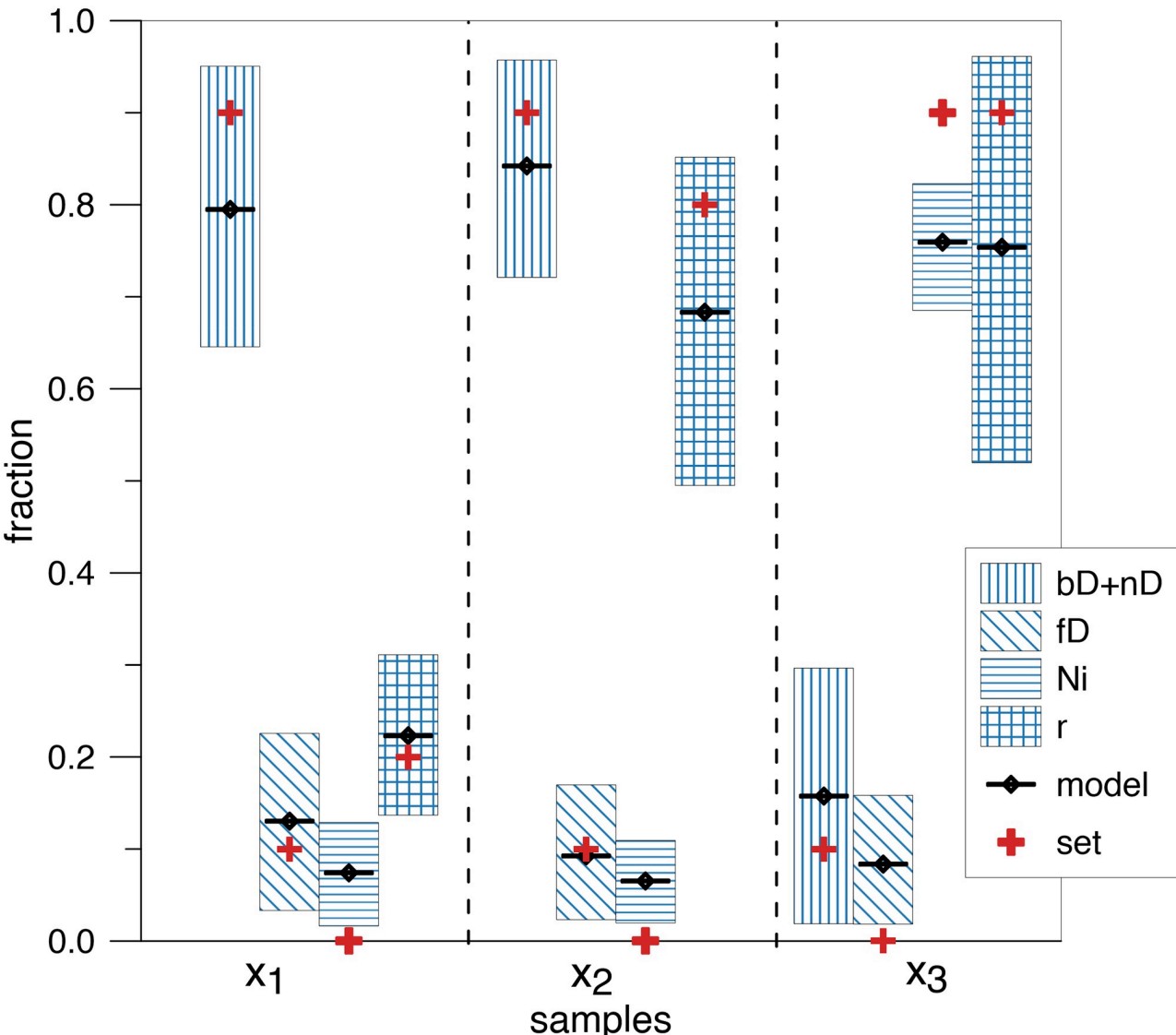

**Fig 12. Comparison of model results (black symbols) with assumed values for three analysed samples (set values—Red crosses).** The bars represent the 68% confidence interval for each fraction. bD and nD fractions are presented jointly due to weak isotopic separation between them.

into the mathematical algorithm. Besides the case studies presented here FRAME can be applied to various other calculations. It can also be potentially applied to other environmental tracers undergoing mixing and possible further alternations, such as compound concentrations, chemical markers or radioactive tracers.

## Supporting information

**S1 Appendix. Supplementary material to the manuscript including the detailed description of the mathematical model, the user guide for the graphical interface as well as a couple of demonstration of interesting use cases.**
(PDF)

## Author Contributions

**Conceptualization:** Maciej P. Lewicki, Dominika Lewicka-Szczebak, Grzegorz Skrzypek.

**Data curation:** Maciej P. Lewicki.

**Formal analysis:** Maciej P. Lewicki.

**Funding acquisition:** Dominika Lewicka-Szczebak.

**Investigation:** Maciej P. Lewicki.

**Methodology:** Maciej P. Lewicki, Dominika Lewicka-Szczebak, Grzegorz Skrzypek.

**Resources:** Maciej P. Lewicki.

**Software:** Maciej P. Lewicki.

**Supervision:** Dominika Lewicka-Szczebak, Grzegorz Skrzypek.

**Validation:** Maciej P. Lewicki.

**Visualization:** Maciej P. Lewicki.

**Writing – original draft:** Maciej P. Lewicki, Dominika Lewicka-Szczebak.

**Writing – review & editing:** Maciej P. Lewicki, Dominika Lewicka-Szczebak, Grzegorz Skrzypek.

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
