## [Decision Letter · Decision Letter 0]

19 Aug 2022

PONE-D-21-35676

FRAME - Monte Carlo model for evaluation of the stable isotope mixing and fractionation

PLOS ONE

Dear Dr. Lewicki,

Thank you for submitting your manuscript to PLOS ONE. After careful consideration, we feel that it has merit but does not fully meet PLOS ONE’s publication criteria as it currently stands. Therefore, we invite you to submit a revised version of the manuscript that addresses the points raised during the review process.

We look forward to receiving your revised manuscript.

Kind regards,

Matheus C. Carvalho

Academic Editor

PLOS ONE

Journal Requirements:

2. Please upload a new copy of Figure 14 as the detail is not clear. Please follow the link for more information: https://blogs.plos.org/plos/2019/06/looking-good-tips-for-creating-your-plos-figures-graphics/" https://blogs.plos.org/plos/2019/06/looking-good-tips-for-creating-your-plos-figures-graphics/

3. Please ensure that you refer to Figures 9, 14, 15, 16, 17, 18, 19, 20, 21, 22, and 23 in your text as, if accepted, production will need this reference to link the reader to the figure.

4. We note that Figures 5 and 14 in your submission contain copyrighted images. All PLOS content is published under the Creative Commons Attribution License (CC BY 4.0), which means that the manuscript, images, and Supporting Information files will be freely available online, and any third party is permitted to access, download, copy, distribute, and use these materials in any way, even commercially, with proper attribution. For more information, see our copyright guidelines: http://journals.plos.org/plosone/s/licenses-and-copyright.

a. You may seek permission from the original copyright holder of Figures 5 and 14 to publish the content specifically under the CC BY 4.0 license. 

5. We note you have included a table to which you do not refer in the text of your manuscript. Please ensure that you refer to Tables 1, 2, 3, 4, 6, 7, 8 and 9 in your text; if accepted, production will need this reference to link the reader to the Table.

Reviewers' comments:

Reviewer's Responses to Questions

**Comments to the Author**

1. Is the manuscript technically sound, and do the data support the conclusions?

Reviewer #1: Yes

Reviewer #2: Yes

2. Has the statistical analysis been performed appropriately and rigorously? 

Reviewer #1: Yes

Reviewer #2: Yes

3. Have the authors made all data underlying the findings in their manuscript fully available?

Reviewer #1: Yes

Reviewer #2: Yes

4. Is the manuscript presented in an intelligible fashion and written in standard English?

Reviewer #1: No

Reviewer #2: Yes

5. Review Comments to the Author

Reviewer #1: General comments

This paper describes a computational model for unraveling mixing and isotopic fractionation that occurs various environmental/geochemical/ecological settings. Since stable isotope ratios often show distinct values between different source materials, isotope ratios of a sample and its potential sources are compared to estimate contribution from each source. However, it is not always the case that proportions of the sources are algebraically calculated because of too many potential sources or isotopic change (fractionation) during chemical/physical conversion of target compounds after mixing. In addition, error associated with the source partitioning is not easy to derive. The authors developed a convenient and flexible tool that can calculate mixing rate of sources with probability distributions and effect of other factors such as isotopic fractionation. They also present many examples showing how the tool can resolve the mixing problems of real conditions appropriately.

Although I cannot fully evaluate the mathematical theory part because it is outside my area of expertise, I think this paper will bring great benefits to researchers using stable isotopes to elucidate material cycles in a variety of disciplines. I recommend its publication after addressing following points.

Specific comments

Abstract. The authors write “this tool can also be useful for non-stable isotope studies”, but no example of such application is listed. It would be better to add some ideas.

L17–18. I think there was an attempt at estimation of errors in source partitioning with consideration of isotopic fractionation although the approach was simpler than the present study (Toyoda, S., Yano, M., Nishimura, S., Akiyama, H., Hayakawa, A., Koba, K., et al. (2011). Characterization and production and consumption processes of N2O emitted from temperate agricultural soils determined via isotopomer ratio analysis. Global Biogeochem. Cycles, 25, GB2008).

L33–43. In the work of Lewicka-Szczebak et al. (2020) they compared two cases: 1) N2O reduction by denitrification first occurs and then mixed with N2O from other sources, 2) N2O reduction occurs after mixing. But in the present paper, it seems only the case 2) is considered. Can the developed tool also simulate the case 1)?

L129. Correct typo (1,m]

L422–425. Although it is not critical in the context, it would be better to note that overall isotope fractionation is not always equal to simple sum of fractionation for the two step in order to avoid misunderstanding.

Figure 1 caption, the last line, “The bottom right panel”. LEFT panel? Also please add titles of axes, same for other figures.

Figure 2. Some characters in the figure do not correspond to those explained in the caption. I think the axis titles “X” and “Y” must be “I1” and “I2”, respectively. Also “M” means x?

Figure 3. I cannot understand why one can tell “the stabilization is achieved” from the figure. To my eyes, oscillating feature is almost the same between before and after the dashed line.

Figures 16–19. It is confusing that caption says the authors consider only three endmembers while five sources are plotted in the figure.

Reviewer #2: The paper presents a working open source framework for stable isotope analisys.

The software is user friendly and shows some original features in respect to other similar works. In particular it can consider fractination process besides isotope mixing.

The paper is quite well written and clear. There are just some minor notes and suggestions for authors:

- The examples are usually well presented and references are always given. However, there can be some difficulties for readers to get a rapid introduction to them. A small introductiion paragraph for each example, what is the addressed problem and what can be gained with isotope analisys, can increase readability.

- The software works flawlessly on Linux/wine, however it is not well clear what is the right combination of data/source/model for each example. Here also an introduction would be welcome. A comment part in the data file would be very useful.

- In the last paragraph of the comment of Figure 1, I think there is an error: right  left.

6. PLOS authors have the option to publish the peer review history of their article (what does this mean?). If published, this will include your full peer review and any attached files.

Reviewer #1: No

Reviewer #2: No

---

## [Author Response · Author response to Decision Letter 0]

29 Sep 2022

Review response

Please see in green our answers to all raised points.

Editor comments:

1.

Please ensure that your manuscript meets PLOS ONE's style requirements, including

those for file naming. The PLOS ONE style templates can be found at

https://journals.plos.org/plosone/s/file?id=wjVg/PLOSOne_formatting_sample_main_body.

pdf and

https://journals.plos.org/plosone/s/file?id=ba62/PLOSOne_formatting_sample_title_author

s_affiliations.pdf

We have reviewed the style guidelines, as well as the file names and all seems to be in order,

following journal requirements. Please let us know if we are missing some specific

formatting.

2. Please upload a new copy of Figure 14 as the detail is not clear. Please follow the link for

more information: https://blogs.plos.org/plos/2019/06/looking-good-tips-for-creating-

your-plos-figures-graphics/" https://blogs.plos.org/plos/2019/06/looking-good-tips-for-

creating-your-plos-figures-graphics/

Indeed the resolution of Figure 14 in the PDF file produced during the manuscript

submission is very poor, similarly the quality of a few other figures is unsatisfactory.

However, the resolution and readability of the original tiff files submitted by us with the

manuscript, are very good . Probably, compression of the figure during PDF production was

reducing the figure resolution too far. Thus, it is not clear if anything can be done on our side

to improve the figure quality. Please let us know if there is anything that we could try to

improve the compressed figure quality.

3. Please ensure that you refer to Figures 9, 14, 15, 16, 17, 18, 19, 20, 21, 22, and 23 in your

text as, if accepted, production will need this reference to link the reader to the figure.

This has been corrected; each figure is now referred to in the text.

4. We note that Figures 5 and 14 in your submission contain copyrighted images. All PLOS

content is published under the Creative Commons Attribution License (CC BY 4.0), which

means that the manuscript, images, and Supporting Information files will be freely available

online, and any third party is permitted to access, download, copy, distribute, and use these

materials in any way, even commercially, with proper attribution. For more information, see

our copyright guidelines: http://journals.plos.org/plosone/s/licenses-and-copyright.

Figures 5 and 14 have been generated using our software, which has been created by the

authors of this manuscript. This software is an original contribution and it is not subject to a

copy rights of third parties.

5. We note you have included a table to which you do not refer in the text of your

manuscript. Please ensure that you refer to Tables 1, 2, 3, 4, 6, 7, 8 and 9 in your text; if

accepted, production will need this reference to link the reader to the Table.This has been corrected; each figure is now referred to in the text.

6. Please review your reference list to ensure that it is complete and correct. If you have

cited papers that have been retracted, please include the rationale for doing so in the

manuscript text, or remove these references and replace them with relevant current

references. Any changes to the reference list should be mentioned in the rebuttal letter that

accompanies your revised manuscript. If you need to cite a retracted article, indicate the

article’s retracted status in the References list and also include a citation and full reference

for the retraction notice.

We have revised our reference list; it is complete and correct.

Reviewers' comments:

Reviewer #1: General comments

This paper describes a computational model for unraveling mixing and isotopic

fractionation that occurs various environmental/geochemical/ecological settings. Since

stable isotope ratios often show distinct values between different source materials, isotope

ratios of a sample and its potential sources are compared to estimate contribution from each

source. However, it is not always the case that proportions of the sources are algebraically

calculated because of too many potential sources or isotopic change (fractionation) during

chemical/physical conversion of target compounds after mixing. In addition, error

associated with the source partitioning is not easy to derive. The authors developed a

convenient and flexible tool that can calculate mixing rate of sources with probability

distributions and effect of other factors such as isotopic fractionation. They also present

many examples showing how the tool can resolve the mixing problems of real conditions

appropriately.

Although I cannot fully evaluate the mathematical theory part because it is outside my area

of expertise, I think this paper will bring great benefits to researchers using stable isotopes

to elucidate material cycles in a variety of disciplines. I recommend its publication after

addressing following points.

Thank you very much for your time devoted to reviewing our manuscript and for your

positive opinion and important suggestions. We have corrected the manuscript according to

your comments, please find our response to the specific points below.

Specific comments

Abstract. The authors write “this tool can also be useful for non-stable isotope studies”, but

no example of such application is listed. It would be better to add some ideas.

Thank you for this comment. Our intention was to indicate that this presented tool may also

be potentially used for solving other mixing problems besides stable isotope studies.

However, since the authors specialise in this area, we have only suitable examples from our

area of expertise, and so far are not able to add examples from non-stable isotope studies.

Therefore, we admit that this statement should be removed from the abstract; we only

indicate this potential application in the outlook and conclusions section.L17–18. I think there was an attempt at estimation of errors in source partitioning with

consideration of isotopic fractionation although the approach was simpler than the present

study (Toyoda, S., Yano, M., Nishimura, S., Akiyama, H., Hayakawa, A., Koba, K., et al. (2011).

Characterization and production and consumption processes of N2O emitted from

temperate agricultural soils determined via isotopomer ratio analysis. Global Biogeochem.

Cycles, 25, GB2008).

Thank you very much for this suggestion. We regret omitting this important paper in our

literature review. Indeed, this study was the first in N2O studies, where estimation of errors

was taken into account. This reference with the relevant information was added to the

introduction (in the paragraph regarding N2O modelling) lines: 44-48.

L33–43. In the work of Lewicka-Szczebak et al. (2020) they compared two cases: 1) N2O

reduction by denitrification first occurs and then mixed with N2O from other sources, 2)

N2O reduction occurs after mixing. But in the present paper, it seems only the case 2) is

considered. Can the developed tool also simulate the case 1)?

Thank you very much for this question; indeed, we missed to explain this in the manuscript.

Yes, the tool can simulate both cases, this can be done by entering the proper equation

describing the fractionation process.

For reduction first (case 1)

M = f bD (S bD +Aln(r)) + f nD S nD + f f D S fD + f Ni S Ni

For reduction after mixing (case 2)

M = f bD S bD + f nD S nD + f fD S fD + f Ni S Ni + Aln(r)

Here, for simplicity, we described case 2 only for N2O. The same calculation algorithm was

used in the previous paper, but now we build it into a software with a graphical interface

improving also propagation of errors and redesigning method for accounting the range of

values for the defined sources.

We agree that to be precise and provide full information it is important to inform readers

that we can take into account the fractionation of the whole mixture as well the fractionation

of only one source before mixing. This information has been added in Section 3.3, where

various types of possible fractionation with respective equations are discussed.

L129. Correct typo (1,m]

This typo has been corrected.

L422–425. Although it is not critical in the context, it would be better to note that overall

isotope fractionation is not always equal to simple sum of fractionation for the two step inorder to avoid misunderstanding.

This is correct. This description was very simplified and actually not needed here because,

as the reviewer suggests, it is not particularly relevant in this context. We further revised

this description and gave only a summary fractionation for the entire process to avoid

misunderstanding.

Figure 1 caption, the last line, “The bottom right panel”. LEFT panel?

Thank you very much for your careful reading! This mistake has been corrected.

Also please add titles of axes, same for other figures.

In the standard FRAME output plots showing the distributions and correlations of variables

estimated in the model, the horizontal axes represent fractions and vertical axes represent

either fractions (in case of correlations) or counts (in case of histograms). After testing

multiple solutions, we decided not to include axis descriptions on these plots, because they

significantly reduce readability. Instead, a careful description of the axes was moved to the

figure caption. Moreover, as these plots are automatically generated by the FRAME software,

it would be very challenging to implement a “one-fit-all” solution for axis description.

Figure 2. Some characters in the figure do not correspond to those explained in the caption. I

think the axis titles “X” and “Y” must be “I1” and “I2”, respectively. Also “M” means x?

The labels in the plot were misleading. The figure has been changed accordingly.

Figure 3. I cannot understand why one can tell “the stabilization is achieved” from the figure.

To my eyes, oscillating feature is almost the same between before and after the dashed line.

A following explanation was added to the figure caption:

In this example, the simulation begins with fractions already close to the equilibrium value,

thus the stabilization is achieved virtually instantly and no visible change in oscillations is

observed in dependence on the iteration number.

Figures 16–19. It is confusing that caption says the authors consider only three endmembers

while five sources are plotted in the figure.

This mistake has been corrected. Thank you for remarking on this.

Reviewer #2: The paper presents a working open source framework for stable isotope

analisys.

The software is user friendly and shows some original features in respect to other similar

works. In particular it can consider fractination process besides isotope mixing.The paper is quite well written and clear. There are just some minor notes and suggestions

for authors:

Thank you very much for your time devoted to reviewing our manuscript and for your

positive feedback and helpful suggestions. Please find our responses to your detailed

comments below.

- The examples are usually well presented and references are always given. However, there

can be some difficulties for readers to get a rapid introduction to them. A small introductiion

paragraph for each example, what is the addressed problem and what can be gained with

isotope analisys, can increase readability.

We have rewritten the example case studies paragraphs, expanding a bit the general

introduction to each case to improve clarity.

- The software works flawlessly on Linux/wine, however it is not well clear what is the right

combination of data/source/model for each example. Here also an introduction would be

welcome. A comment part in the data file would be very useful.

We have added a description of each case in the supplement (appendix B) with an indication

of which files should be applied for which case.

- In the last paragraph of the comment of Figure 1, I think there is an error: right  left.

Yes, thank you for your careful reading! This mistake has been corrected.

---

## [Decision Letter · Decision Letter 1]

17 Oct 2022

PONE-D-21-35676R1FRAME - Monte Carlo model for evaluation of the stable isotope mixing and fractionationPLOS ONE

Dear Dr. Lewicki,

Thank you for submitting your manuscript to PLOS ONE. After careful consideration, we feel that it has merit but does not fully meet PLOS ONE’s publication criteria as it currently stands. Therefore, we invite you to submit a revised version of the manuscript that addresses the points raised during the review process. Please submit your revised manuscript by Dec 01 2022 11:59PM. If you will need more time than this to complete your revisions, please reply to this message or contact the journal office at plosone@plos.org. Please include the following items when submitting your revised manuscript:A rebuttal letter that responds to each point raised by the academic editor and reviewer(s). You should upload this letter as a separate file labeled 'Response to Reviewers'.A marked-up copy of your manuscript that highlights changes made to the original version. You should upload this as a separate file labeled 'Revised Manuscript with Track Changes'.An unmarked version of your revised paper without tracked changes. You should upload this as a separate file labeled 'Manuscript'.If applicable, we recommend that you deposit your laboratory protocols in protocols.io to enhance the reproducibility of your results. Protocols.io assigns your protocol its own identifier (DOI) so that it can be cited independently in the future. For instructions see: https://journals.plos.org/plosone/s/submission-guidelines#loc-laboratory-protocols. Additionally, PLOS ONE offers an option for publishing peer-reviewed Lab Protocol articles, which describe protocols hosted on protocols.io. Read more information on sharing protocols at https://plos.org/protocols?utm_medium=editorial-email&utm_source=authorletters&utm_campaign=protocols.

We look forward to receiving your revised manuscript.

Kind regards,

Viacheslav Kovtun, Dr.Sc., Ph.D.

Academic Editor

PLOS ONE

Journal Requirements:

Reviewers' comments:

Reviewer's Responses to Questions

**Comments to the Author**

1. If the authors have adequately addressed your comments raised in a previous round of review and you feel that this manuscript is now acceptable for publication, you may indicate that here to bypass the “Comments to the Author” section, enter your conflict of interest statement in the “Confidential to Editor” section, and submit your "Accept" recommendation.

Reviewer #1: (No Response)

Reviewer #2: All comments have been addressed

2. Is the manuscript technically sound, and do the data support the conclusions?

Reviewer #1: Yes

Reviewer #2: Yes

3. Has the statistical analysis been performed appropriately and rigorously? 

Reviewer #1: N/A

Reviewer #2: N/A

4. Have the authors made all data underlying the findings in their manuscript fully available?

Reviewer #1: Yes

Reviewer #2: Yes

5. Is the manuscript presented in an intelligible fashion and written in standard English?

Reviewer #1: Yes

Reviewer #2: Yes

6. Review Comments to the Author

Reviewer #1: In the revised manuscript, the problems I pointed out have been improved. I think it is now acceptable after a few minor corrections below.

L12–13. I cannot understand why the authors revised this phrasing while they did not change the words at L4. I cannot make sense out of the phrase "n elements stable isotope composition determination"

L50. Typo. “signature”

Reviewer #2: (No Response)

7. PLOS authors have the option to publish the peer review history of their article (what does this mean?). If published, this will include your full peer review and any attached files.

Reviewer #1: No

Reviewer #2: No

---

## [Author Response · Author response to Decision Letter 1]

20 Oct 2022

Dear Dr Kovtun,

We are pleased to resubmit after the second review process our manuscript (PONE-D-21-35676 )

entitled “ FRAME - Monte Carlo model for evaluation of the stable isotope mixing and fractionation”

after additional few minor corrections for consideration in PlosONE journal.

Please see our responses in green below the reviewers’ comments and the changes marked in the

PDF file. The two corrections pointed out by the reviewer 1 are purely editorial and have been

included in the manuscript.

We look forward to the acceptance of our manuscript.

Sincerely yours,

Maciej Lewicki

Corresponding author on behalf of all authors

Review response

Reviewer #1: 

In the revised manuscript, the problems I pointed out have been improved. I think it is

now acceptable after a few minor corrections below.

> Thank you very much for checking our manuscript again. We have corrected the

> mistakes you pointed out as requested.

L12–13. I cannot understand why the authors revised this phrasing while they did not

change the words at L4. I cannot make sense out of the phrase "n elements stable

isotope composition determination"

> This is right, this phrasing was too complex, we changed this to use consequently ‘n

> isotopes analysed in a sample’

L50. Typo. “signature”

> It was corrected.

---

## [Decision Letter · Decision Letter 2]

24 Oct 2022

FRAME - Monte Carlo model for evaluation of the stable isotope mixing and fractionation

PONE-D-21-35676R2

Dear Dr. Lewicki,

We’re pleased to inform you that your manuscript has been judged scientifically suitable for publication and will be formally accepted for publication once it meets all outstanding technical requirements.

Kind regards,

Viacheslav Kovtun, Dr.Sc., Ph.D.

Academic Editor

PLOS ONE

Additional Editor Comments (optional):

Reviewers' comments:

Reviewer's Responses to Questions

**Comments to the Author**

1. If the authors have adequately addressed your comments raised in a previous round of review and you feel that this manuscript is now acceptable for publication, you may indicate that here to bypass the “Comments to the Author” section, enter your conflict of interest statement in the “Confidential to Editor” section, and submit your "Accept" recommendation.

Reviewer #1: All comments have been addressed

Reviewer #2: All comments have been addressed

2. Is the manuscript technically sound, and do the data support the conclusions?

Reviewer #1: Yes

Reviewer #2: Yes

3. Has the statistical analysis been performed appropriately and rigorously? 

Reviewer #1: Yes

Reviewer #2: N/A

4. Have the authors made all data underlying the findings in their manuscript fully available?

Reviewer #1: Yes

Reviewer #2: Yes

5. Is the manuscript presented in an intelligible fashion and written in standard English?

Reviewer #1: Yes

Reviewer #2: Yes

6. Review Comments to the Author

Reviewer #1: (No Response)

Reviewer #2: All comments have been addressed.

7. PLOS authors have the option to publish the peer review history of their article (what does this mean?). If published, this will include your full peer review and any attached files.

Reviewer #1: No

Reviewer #2: No

---

## [Editor Report · Acceptance letter]

8 Nov 2022

PONE-D-21-35676R2 

FRAME - Monte Carlo model for evaluation of the stable isotope mixing and fractionation 

Dear Dr. Lewicki:

I'm pleased to inform you that your manuscript has been deemed suitable for publication in PLOS ONE. Congratulations! Your manuscript is now with our production department. 

Kind regards, 

on behalf of

Professor Viacheslav Kovtun 

Academic Editor

PLOS ONE